# FBP2—A New Player in Regulation of Motility of Mitochondria and Stability of Microtubules in Cardiomyocytes

**DOI:** 10.3390/cells11101710

**Published:** 2022-05-21

**Authors:** Łukasz Pietras, Ewa Stefanik, Dariusz Rakus, Agnieszka Gizak

**Affiliations:** Department of Molecular Physiology and Neurobiology, University of Wrocław, Sienkiewicza 21, 50-335 Wrocław, Poland; lukasz.pietras2@uwr.edu.pl (Ł.P.); 300359@uwr.edu.pl (E.S.); dariusz.rakus@uwr.edu.pl (D.R.)

**Keywords:** FBP2, mitochondrial motility, microtubule stability, HL-1 cardiomyocytes, Tau, MAP1B

## Abstract

Recently, we have shown that the physiological roles of a multifunctional protein fructose 1,6-bisphosphatase 2 (FBP2, also called muscle FBP) depend on the oligomeric state of the protein. Here, we present several lines of evidence that in HL-1 cardiomyocytes, a forced, chemically induced reduction in the FBP2 dimer-tetramer ratio that imitates AMP and NAD^+^ action and restricts FBP2-mitochondria interaction, results in an increase in Tau phosphorylation, augmentation of FBP2-Tau and FBP2-MAP1B interactions, disturbance of tubulin network, marked reduction in the speed of mitochondrial trafficking and increase in mitophagy. These results not only highlight the significance of oligomerization for the regulation of FBP2 physiological role in the cell, but they also demonstrate a novel, important cellular function of this multitasking protein—a function that might be crucial for processes that take place during physiological and pathological cardiac remodeling, and during the onset of diseases which are rooted in the destabilization of MT and/or mitochondrial network dynamics.

## 1. Introduction

In the last 10 years, fructose 1,6-bisphosphatase (FBP, EC 3.1.3.11) has become a hot topic among enzymes of energy metabolism. This is because it has been shown that FBP may regulate not only glucose/glycogen synthesis from precursors of carbohydrates but it may also influence—via interactions with a set of proteins (e.g., mitochondrial VDAC, ANT, and ATP synthase, CAMK2, and transcription factors HIF1α and NF-κB)—cell cycle-dependent events, mitochondria biogenesis and polarization of their membranes, expression of glycolytic enzymes, induction of synaptic plasticity and even cancer progression [1,2,3,4,5,6].

In mammalian tissues, two different FBP isozymes are expressed: the liver FBP (FBP1) and the muscle (FBP2) isozyme. They catalyze the hydrolysis of fructose-1,6-phosphate to fructose-6-phosphate and inorganic phosphate. Both isozymes form homotetramers. In the presence of their allosteric inhibitors—AMP and NAD^+^, FBP1 and FBP2 tetramers adopt a similar, inactive T-state in which two upper subunits (the upper dimer) are slightly rotated in respect to the lower two subunits (the lower dimer). In the absence of the inhibitors, tetramers of both isozymes adopt the active R-state. However, while the FBP1 tetramer is almost flat (for review see [7]), FBP2 adopts a unique cross-like structure in which the upper dimer is twisted about 90° with respect to the lower one [8]. FBP2 may also exist as a dimer that is fully active and is not inhibited by AMP and NAD^+^ since the mechanism of the allosteric inhibition requires the presence of the tetrameric conformation: binding of AMP to subunits within one dimer inhibits the catalytic sites in the second dimer. Importantly, both in the unique R-state of FBP2 and in its dimeric form, additional surfaces are exposed to the solution and hence, they may form new (as compared to FBP1 and also to the T-state of FBP2 tetramer) docking sites for binding partners (for review see [7]).

In HL-1 cardiomyocytes and KLN205 squamous cell carcinoma cell lines, partial silencing of the muscle isozyme of FBP expression leads to sensitization of mitochondria to membrane-depolarizing stress conditions [1,3]. Although we have found that induction of FBP2 accumulation on the organelles by such conditions (e.g., by high [Ca^2+^], high ROS or inhibition of GKS3 activity) reduces mitochondrial swelling and correlates with the fragmentation of the mitochondrial network, we have also shown that a fraction of the enzyme interacts with less fragmented, filamentous mitochondria in control conditions [1]. This suggests that the enzyme is not directly engaged in the process of mitochondrial division, at least in basal conditions.

The physiological function of FBP2 is dependent not only on the level of its expression in a cell but also on its oligomeric state. The dimeric form associates with mitochondria and protects them against stress stimuli, the tetrameric form of FBP2 is retained in the cell nucleus [9]. Additionally, in co-cultures of cardiomyocytes and fibroblasts, as well as neurons and astrocytes, the level of cellular FBP2 protein and its localization is different than in monocultures of these cells, which is regulated by factors released to the culture medium in extracellular vesicles (EVs) [10]. It thus appears that modulation of the oligomeric state of FBP2 (by low molecular weight compounds or other factors) changes the set of cellular binding partners of FBP2 and enables the switching of its action from the promotion of cell survival to induction of death [9].

Most recently, we have found that a novel remitting leukodystrophy was associated with a Val115Met variant of FBP2 [11]. In fibroblasts carrying the variant, FBP2 was unable to co-localize with mitochondria which correlated with a disturbance of the mitochondrial network and an increase in ROS production. In particular, in the majority of the cells, mitochondria appeared to aggregate in the central region of cells, rather than form extended networks as in healthy cells. This aggregation could be a result of the diminished motility of mitochondria along microtubules. This, in turn, might suggest that the perturbation of the Val115Met FBP2-mitochondria interaction contributed to the changes in the motility of the organelles. In healthy cells, such an interaction might be disturbed by the induction of tetramerization of FBP2 (e.g., by its physiological effectors—AMP or NAD^+^) [9].

Therefore, we sought to clarify the contribution of FBP2 to this aggregation of mitochondria in HL-1 cardiomyocytes—the cells that express only the FBP2 isozyme, and in which the pathways regulating the intracellular behavior of the protein were best studied. We found that chemically induced tetramerization of FBP2 resulting in reduction in FBP2-mitochondria interaction correlated with a disturbance of tubulin network, marked reduction in mitochondrial mobility and increase in mitophagy. In turn, the induction of FBP2-mitochondria interactions by EVs isolated from cardiac fibroblasts correlated with an increase in mitochondrial mobility.

It therefore appears that FBP2 is a new player in the regulation of microtubule stability-dependent trafficking of mitochondria in cardiomyocytes.

## 2. Materials and Methods

### 2.1. Cell Culture

The immortalized and dividing HL-1 cardiomyocytes cell line [12] was a gift from Dr. W.C. Claycomb (Louisiana State University Health Science Center, New Orleans, LA, USA). They were cultured in Claycomb medium, supplemented with 10% HL-1 cells screened fetal bovine serum (cat. no TMS-016), 2 mm L-glutamine, 0.1 mm noradrenalin and penicillin/streptomycin (all the chemicals supplied by Merck, Darmstadt, Germany) as described before [13].

Murine cardiac fibroblasts were isolated from the hearts of newborn C57BL/6 mice according to [14]. The protocol complies with standards of EU Directive 2010/63/EU for animal experiments and was approved by the II Local Scientific Research Ethical Committee, Wroclaw University of Environmental and Life Sciences (permission no WNB.464.2.2020.IR). The fibroblasts were cultured in a DMEM medium with supplements as described before [10].

### 2.2. HL-1 Cardiomyocytes Treatment with Fibroblasts-Derived Extracellular Vesicles (EVs)

After 48 h of fibroblast culture, the culture medium from a 25 mL culture flask was collected and extracellular vesicles were purified according to the method described in [15]. HL-1 cells were transfected using CellLight™ Mitochondria-GFP, BacMam 2.0 (ThermoFisher Scientific, Waltham, MA, USA) and incubated with the vesicles for 24 h. Alternatively, the cells were preincubated with 10 µM forskolin (Merck, Darmstadt, Germany) before application of EVs or treated with 2 µM PKA inhibitor (KT5720, Merck, Darmstadt, Germany) for 24 h. The velocity of at least 144 mitochondria was measured for each treatment. The experiment was performed in duplicate.

To test the effect of PKA inhibitor on the ability of nocodazole to destabilize microtubules, HL-1 cells were pretreated with a 2 µM PKA inhibitor for 15 min and then incubated with 0.1 µM nocodazole (Merck, Darmstadt, Germany) for 1 h. Cells were fixed, and microtubules were stained as described in the Immunofluorescence subsection.

### 2.3. Tetramerization of FBP2

To tetramerize FBP2 [16] HL-1 cardiomyocytes were treated with 5 µM FBP2 allosteric inhibitor (5-chloro-2-(N-(2,5-dichlorobenzenesulfonamido))-benzoxazole; Cayman Chemicals, Ann Arbor, MI, USA) [17,18], hereinafter referred to as iFBP2, for 1 h prior to further experiments.

### 2.4. Silencing of FBP2 Expression

FBP2 expression was silenced with muscle FBPase shRNA Lentiviral Particles (Santa Cruz Biotechnology, Dallas, TX, USA) according to the manufacturer’s protocol. A stable line of the shRNA expressing HL-1 cells was established using puromycin dihydrochloride (15 µg/mL). The silencing was monitored by FISH, immunofluorescence and FBP2 enzymatic activity measurement.

### 2.5. Measurement of FBP2 Activity

FBP2 enzymatic activity was measured using the glucose 6-phosphate isomerase—glucose 6-phosphate dehydrogenase coupled assay. The reduction of nicotinamide adenine dinucleotide phosphate (NADP^+^) to NADPH was monitored spectrophotometrically at 340 nm [19]. The enzyme activity was expressed in U [mol min^−1^].

### 2.6. Fluorescent in Situ Hybridization (FISH)

FISH was performed as described previously [20]. The Fbp2 oligonucleotide was synthesized by Merck, Darmstadt, Germany, and has the sequence (5′-[Cyanine3]GCACACAGCT GAGATACTCT TGCACATCCT CAGGGGAC-3′). In the control reaction, the oligonucleotide probe was omitted. The experiment was performed in triplicate.

### 2.7. Immunofluorescence

Cells growing on coverslips were fixed in 4% paraformaldehyde, permeabilized with 0.1% Triton X-100 in PBS and incubated with 3% BSA in PBS to reduce nonspecific binding of antibodies (all the chemicals supplied by Merck, Darmstadt, Germany). Then they were incubated overnight with rabbit anti-FBP2 antibody [13], and for 1 h with anti-rabbit Alexa 633-conjugated secondary antibodies (A-21070; ThermoFisher Scientific, Waltham, MA, USA). To stain microtubules the cells were incubated with mouse anti-α tubulin antibody FITC-conjugated (F2168; Merck, Darmstadt, Germany). To test the FBP2-mitochondria co-localization, murine anti-FBP2 antibody [13], rabbit anti-TOMM antibody (HPA011562; Merck, Darmstadt, Germany) and appropriate secondary antibodies Alexa 633- or FITC-conjugated (ThermoFisher Scientific or Merck, respectively) were used. For the analysis of FBP2 co-localization with mitochondria and microtubules with mitochondria, the Manders’ coefficient (M) was determined using the JACoP plugin of the ImageJ/FIJI as we described before [11]. The coefficient varies from 0 (no co-localization) to 1 (100% of co-localization). For the measurements, only the fluorescent signal from the cytoplasmic area was taken into account, the nuclear signal was excluded.

FBP2-protein interactions were assayed using the DuoLink^®^ Proximity Ligation Assay (DuoLink^®^ In Situ Orange Starter Kit Mouse/Rabbit; Merck, Darmstadt, Germany) according to the vendor’s instruction, and anti-phospho-Tau (pThr231) (SAB450456; Merck, Darmstadt, Germany), anti-Tau (ZMS3420; Merck, Darmstadt, Germany), anti-MAP1B (sc-365668; Santa Cruz Biotechnology, Dallas, TX, USA), and appropriate (rabbit or mouse) anti-FBP2 antibodies. In control reactions, primary antibodies were omitted. The cells were immersed in Fluoroshield with DAPI (Merck, Darmstadt, Germany) and mounted on microscopic slides. The experiments were performed in triplicate. Fluorescence from at least 300 cells was measured in each condition.

### 2.8. Confocal Microscopy

Images were acquired on the FV-1000 confocal microscope (Olympus, Tokyo, Japan) with 60× (oil, Plan SApo, NA = 1.35) objective using the Sequential Scan option. The edges of the cells were marked, then the fluorescence intensity was measured and the corrected total cell fluorescence (CTCF) of individual cells was calculated using the ImageJ/FIJI open software (http://imagej.nih.gov/ij/) accessed on 15 January 2022 [21].

### 2.9. Mitochondria Motility and Shape Measurements

Mitochondria were visualized using CellLight™ Mitochondria-GFP, BacMam 2.0 (ThermoFisher Scientific, Waltham, MA, USA). To visualize mitochondrial movement in living cells, the HL-1 cells were cultured in NuncTM Glass Base Dish, 12 mm (ThermoFisher Scientific, Waltham, MA, USA). The HL-1 cells were incubated with a studied factor for 1 h (5 µM iFBP2, 10 µM GSK3 inhibitor SB216763, 1 µM PI3K inhibitor wortmannin, 5.5 mm glucose; Merck, Darmstadt, Germany) and images were acquired using the Time Scan option the FV-1000 confocal microscope. Time-lapse videos were created on 3× digital zoom and analyzed using FV-10-ASW 4.2 Viewer software (Olympus). To avoid the measurement of random movement of mitochondria powered by cytoplasm fluctuations only the organelles that moved at least 2 μm during the image acquisition time were taken into account [22]. If a mitochondrion moved with pausing episodes, the mean velocity was counted rejecting pause time. The experiments were replicated at least three times, and velocities of at least 482 mitochondria were measured for each treatment.

The length and shape of mitochondria were analyzed using the MiNA (Mitochondrial Network Analysis by Sturat Lab) plugin of the ImageJ/FIJI. At least 5480 mitochondria were measured in each condition.

### 2.10. Mitochondrial Membrane Potential and Mitophagy

Mitochondrial membrane potential was measured with JC-1 dye (ThermoFisher Scientific, Waltham, MA, USA) as described in [11]. Polarized mitochondria that accumulated more JC-1 dye were red. The reduction in the polarization was reflected by the increase in green fluorescence. The ratio of red to green fluorescence reflected the degree of the mitochondrial membrane polarization. The measurements were taken from at least 250 cells. For a positive control, the mitochondrial membrane was depolarized with 5 µM carbonyl cyanide m-chlorophenyl hydrazone (CCCP), a mitochondrial uncoupling reagent.

The intensity of mitophagy in HL-1 cells with partially silenced FBP2 expression or treated with the 5 µM FBP2 inhibitor was assayed using the Mitophagy Detection Kit (Dojindo Laboratories, Kumamoto, Japan) according to the manufacturer’s instruction. Briefly, cells were stained with Mtphagy Dye. The dye was immobilized in intact mitochondria and exhibited a weak fluorescence. When damaged mitochondria fused to lysosomes their fluorescence emission markedly increased. In a “positive control”, cells were treated with the mitophagy-stimulating concentration of CCCP (10 µm). Cells were observed in confocal microscopy and the intensity of fluorescence was measured as described above (“Confocal microscopy”). The measurements were taken from at least 400 cells.

### 2.11. Scratch Assay

A scratch (wound healing) assay was performed according to [23]. Briefly, WT (wild type, control) HL-1 cells were grown to 90% confluence on a 12-well cell culture plate. The “wound line” (cell-free area) was created using a 10 μL micropipette tip and the floating cells were removed. The recolonization of the area was photographed immediately, and 24 and 48 h after wounding. The average width of a gap was calculated from the microscopic images taken at 5 different sites from each wound line. The experiment was performed in triplicate (in total, 9 wound lines for each condition).

### 2.12. Statistics

Data were analyzed using Statistica 13.3 by StatSoft Inc. (Tulsa, OK, USA) and expressed using median or mean (marked as x on the plots) and standard deviation values, depending on the data distribution tested using the Shapiro–Wilk test. For two data sets the significance of differences was tested using the Student’s *t*-test (for data with normal distribution and equal variances) or the Mann–Whitney U test. The Kruskal–Wallis test was used for the comparison of more than two data sets. The probability *p* < 0.05 was considered to be a significant difference. Data are presented as box-and-whiskers plots where the median is marked by a horizontal line, box limits indicate the 25th and 75th percentiles and outliers are represented by dots.

### 2.13. Mass Spectrometry

After 48 h of monoculture of cardiac fibroblasts, the culture medium was collected from a 75 mL culture flask and extracellular vesicles were purified as described before [15]. To find mitochondria-bound FBP2-interacting proteins, affinity chromatography with Sepharose-FBP2 column and isolated mitochondria homogenate were used (described in [1]). The protein contents of the extracellular vesicles and the eluate from the chromatographic column were analyzed commercially in the Mass Spectrometry Laboratory at the Institute of Biochemistry and Biophysics of the Polish Academy of Sciences, Warsaw, Poland. All software used is accessible at http://proteom.ibb.waw.pl/index.en.html (accessed on 25 March 2022).

## 3. Results and Discussion

Mitochondrial network dynamics encompass mitochondrial fusion, fission, and active transport along microtubules and mitophagy (mitochondrial autophagy) and are crucial for the maintenance of the proper quality of these organelles. Thus, disturbance in mitochondrial dynamics is a component of a variety of diseases. In cardiac myocytes, the appropriate intracellular distribution of mitochondria is essential for sequestration and release of Ca^2+^ regulating (together with ER cisterns) its local and global concentration (for review see [24]. The organelles’ trafficking is also responsible for meeting the cellular region-specific energy requirements and for the efficient removal of excess lactate from the cytoplasm. All these factors are crucial for maintaining continuous contractile activity or cardiomyocytes.

### 3.1. The Effect of FBP2 Silencing and Oligomerization on Mitochondrial Network in HL-1 Cardiomyocytes

Previous studies have indicated that the binding of FBP2 dimer to mitochondria had protective functions and that partial silencing of the protein expression or chemically induced tetramerization of FBP2 (limiting FBP2-mitochondria interactions) increased susceptibility to mitochondria-depolarizing high Ca^2+^ and high ROS stress, thus reducing the viability of HL-1 cardiomyocytes [1,9]. The only exception was hypoxic conditions in which partial silencing of FBP2 expression not only did not reduce but even increased the viability of the cells [9]. Under normoxic conditions, in a number of HL-1 cardiomyocytes with partially silenced FBP2 expression (HL-1 FBP2- cells), Mito-tracker-positive spherical aggregates and toroids (“donut” mitochondria) were observed (previously unpublished). Additionally, in fibroblasts of patients with the V115M-FBP2 variant that does not bind to mitochondria, the organelles aggregated around the nucleus rather than creating a network, which correlated with an increased susceptibility of the cells to stress stimuli [11]. This prompted us to search for mechanisms by which FBP2 can contribute to the observed changes in the mitochondrial network.

To this end, we partially silenced FBP2 expression in HL-1 cells (the cells express only the FBP2 isozyme [13]) using shRNA. To confirm the silencing, the level of FBP2 mRNA in HL-1 FBP2- cells were checked using FISH (Figure 1A), and the level of FBP2 protein using immunofluorescence (Figure 1B) and also, indirectly, by measurement of FBP2 activity in homogenates. Results of the FISH experiment suggested a ~40% reduction in FBP2 mRNA after the shRNA treatment. However, we were more interested in the changes at the protein level. The immunofluorescence experiment revealed about a 36% decrease in the intensity of FBP2-related fluorescence. As can be seen in microscopic pictures in Figure 1B, the silencing resulted in a relatively higher reduction of FBP2-related fluorescence in the nucleus than in the cytoplasm. This may be explained by the fact that FBP2 silencing is considered a stress stimulus to the cells and, in contrast to the cell cycle regulation-related [2] tetramer, the stress-protecting dimeric form of FBP2 does not accumulate in the nucleus [1,9].

The FBP2 enzymatic activity in homogenate from HL-1 FBP2- cells decreased to about 40% of the activity of control HL-1 cells (Figure 1C). Together, these results convincingly demonstrated partial silencing of the FBP2 protein expression.

Because we have demonstrated earlier that FBP2 silencing resulted in a significant reduction in mitochondrial membrane polarity [1], we also tested this parameter and obtained the expected results (Figure 2). Alternatively, we treated control HL-1 cells (HL-1 WT, wild-type) with FBP2 inhibitor (iFBP2). This inhibitor mimics the action of physiological effectors of FBP2 (AMP and NAD^+^) and induces tetramerization of the protein in its inactive T-state [16,18] which in turn, reduces the interaction of FBP2 with mitochondria [16]. As expected, this treatment also resulted in decreased polarization of the mitochondrial membrane (Figure 2). In the positive control, CCCP treatment was used. Moreover, although the CCCP-induced depolarization measured as the JC-1 red-to-green fluorescence ratio appeared to be similar to iFBP2-treated cells, the JC-1 monomer-related signal was highly granular suggesting that the majority of the CCCP-treated mitochondrial network was depolarized strongly enough not to accumulate the dye at all.

In summary, both the decrease in the total FBP2 amount by shRNA and the reduction in the dimer-tetramer ratio with iFBP2 effectively limited the cellular pool of FBP2 molecules that could interact with mitochondria and influence their membrane potential.

### 3.2. FBP2 Silencing and Tetramerization Affects Microtubule Network and Mitochondria Motility

Previously, we searched for FBP2 interactors in HL-1 cardiomyocytes, in the context of the influence of FBP2 on mitochondrial membrane polarity [1]. Among the potential FBP2-interacting proteins detected by commercial mass spectrometry (MS) [1] were α and β tubulin subunits as well as MAP1B (Appendix A)—one of the microtubule-associated proteins (MAPs), proteins that regulate MT assembly and stability).

At that time, however, it did not arouse our great interest, as we concentrated on the proteins that comprised mitochondrial permeability transition pores. Microtubules (MT) form transport tracks for organelles, including mitochondria. MAP1B, together with the Tau protein are the main MAPs stabilizing MT structure in neurons, and they also regulate the migration of mitochondria in opposing directions: antero- and retrograde, respectively [25]. Both Tau and MAP1B are also present in the heart, where changes in their expression are correlated with cardiac pathologies [26,27].

Therefore, we decided to measure the rate of mitochondrial motility in HL-1 FBP2- and iFBP2-treated cells. Both treatments resulted in a significant, over 25%, reduction in mitochondria motility (Figure 3A).

Then, we tested the effect of FBP2 tetramerization on MT integrity. A 1-h incubation of HL-1 WT cells with iFBP2 resulted in a change of MT cytoskeleton appearance from ordered and filamentous to less ordered and with shorter filaments (Figure 3B). Partial silencing of FBP2 expression had a much weaker effect (Figure 3B). Most likely, in contrast to the forced tetramerization of FBP2 molecules, partial silencing of FBP2 expression reduced the overall amount of FBP2 protein, but it did not decrease the dimer-tetramer ratio in the cytoplasm and hence, no alterations in the MT cytoskeleton architecture were observed. We cannot completely exclude the possibility that iFBP2 might directly affect the structure of MT, however, such an effect was not observed in a study using hepatoma cells [18].

### 3.3. FBP2 Tetramerization Influences Mitophagy and Shape of Mitochondrial Network

Mitochondrial motility is important not only for the proper positioning of the organelles in the cell but also for the fusion–fission processes which facilitate mitochondria repair and removal of irreversibly damaged mitochondria by autophagy (mitophagy). Thus, reduced motility might lead to an increase in the number of dysfunctional mitochondria and, as a result, to the intensification of mitophagy. Thus, we analyzed the degree of mitophagy in HL-1 FBP2- cells, WT HL-1 cells and iFBP2-treated WT HL-1 cells using the Mitophagy Detection Kit. To obtain a “positive control” we treated the control cells with carbonyl cyanide m-chlorophenyl hydrazone (CCCP), a mitochondrial uncoupler.

Tetramerization of FBP2 appeared to be as effective in stimulating mitophagy as the CCCP treatment. Surprisingly, the partial silencing of FBP2 expression did not stimulate mitophagy (Figure 4).

This might be explained by the different influence of iFBP and partial silencing of FBP2 on the MT cytoskeleton. To shed some more light on this discrepancy, we analyzed the influence of these two conditions on the shape of mitochondria.

The results of the analysis of the mitochondrial shape revealed that in untreated HL-1 WT cells, the organelles were longer and more branched than in the iFBP2-treated cells (Figure 5 and Appendix A). In turn, in the HL-1 FBP2- cells, the organelles were similar in shape/length to that of HL-1 WT cells (Figure 5 and Appendix A), except that the FBP2 partial silencing resulted in the emergence of numerous almost immobile aggregates and donut-shaped mitochondria [28] (Figure 5), structures that rarely were present in other conditions.

Donut-shaped mitochondria have a higher tolerance to an increase in matrix volume, and their formation is stimulated by the partial detachment of mitochondria from the cytoskeleton [28]. Thus, the emergence of donut mitochondria in HL-1 FBP2- cells might explain the observed deceleration of their migration rate despite limited changes in the MT cytoskeleton, and lack of significant increase in mitophagy despite reduced mitochondrial membrane potential.

Our results also highlight the differences between the effects of an externally imposed, drastic change in the FBP2 dimer/tetramer ratio and just a decrease in the amount of available FBP2 molecules that does not significantly affect the ratio in the cell.

Mitochondrial motility and distribution might also depend on actin filaments [29]. Although the majority of the actin cytoskeleton in HL-1 cardiomyocytes has a specific, sarcomeric, organization, to be on the safe side, we decided to check if the disturbance in the dimer/tetramer ratio by iFBP2 can influence the actin cytoskeleton. Not quite unexpectedly, in both untreated and iFBP2-treated cells actin organization in the cell interior was the same (Appendix A). Furthermore, since FBP2 is a glyconeogenic enzyme that catalyzes the reaction opposite to phosphofructokinase (a regulatory enzyme of glycolysis), its tetramerization in the inactive state might disturb the energy balance in the cell. Although the main energy substrate of HL-1 cardiomyocytes is glutamine, we checked if the addition of 5.5 mm glucose to the culture medium might change the effects of FBP2 tetramerization on mitochondrial motility. We found that in the presence of glucose, iFBP2 was as effective in reducing mitochondrial motility as in the absence of glucose (Appendix A).

To summarize this part of our results, tetramerization of FBP2 resulted in a reduction in FBP2-mitochondria interactions which correlated with a decrease in mitochondrial motility, intensification of mitophagy and disturbance of MT cytoskeleton. In contrast, partial silencing of FBP2 expression (that mitigated the amount of available dimers but was not expected to change the cellular dimer/tetramer equilibrium) was sufficient to reduce mitochondrial motility but probably not as an effect of MT disturbance, but rather a partial detachment of mitochondria from MT. Therefore, in subsequent experiments, we focused on the effects of changing the dimer-tetramer ratio in the cell.

### 3.4. Tetramerization of FBP2 Influences Motility of HL-1 Cells

Microtubules are not only transport tracks for organelles, but they also are force generators, support cellular protrusions, and contribute to the sequence of events that allow for directional migration of the cell. Therefore, we asked if the observed disturbance in MT cytoskeleton and mitochondrial network properties/characteristics in iFBP2-treated cells could influence the migratory capacity of HL-1 cells. To find the answer we used the scratch (wound healing) assay. Results of the assay showed that the iFBP2-induced tetramerization of FBP2 correlated with a noticeable reduction in the rate of wound healing (Figure 6).

### 3.5. Tetramerization of FBP2 Influences Interaction of the Protein with MAP1B and Tau, and Reduces Co-Localization of Mitochondria with Microtubules

In search of a molecular basis for the iFBP2 action, we tested the interactions of FBP2 with MAP1B and also Tau. Tau is a phosphoprotein, and its phosphorylation initiates its dissociation from MT. However, the final effect depends on the specific site(s) of the phosphorylation [30]. For example, the phosphorylation of Thr231 has been shown to result in the detachment of Tau from MT abolishing, thus its ability to polymerize MT [30,31]. Moreover, the abnormally phosphorylated (hyperphosphorylated), detached from MT Tau molecules can aggregate or sequester other MT-interacting proteins which may disrupt MT assembly and lead to so-called tauopathies, e.g., Alzheimer’s disease [32]. However, due to the disordered structure of the Tau protein, the precise mechanism of its action as an MT stabilizer is not fully known. Tau is also present in cardiac tissue and changes in its expression or accumulation of hyperphosphorylated forms have been correlated with mitochondrial abnormalities and deterioration in cardiovascular performance [26,33]. Using immunocytochemistry, we found that 1h treatment of HL-1 WT cells with iFBP2 correlates with a marked (~27%) increase in fluorescence related to Thr231-phosphorylated Tau (pTau; Figure 7). In the same time, the total Tau-related fluorescence did not increase (Figure 7).

Since anti-total Tau antibodies recognize both phosphorylated and unphosphorylated pools of Tau proteins, these results suggested that the expression of Tau was largely unchanged, only phosphorylation of the protein was elevated under iFBP2 treatment.

Moreover, surprisingly, results of the DuoLink^®^ Proximity Ligation assay using anti-FBP2 and anti-total Tau or anti-pTau antibodies suggested that some FBP2 molecules remained in close proximity to Tau in control conditions and that FBP2 tetramerization further induced interactions between FBP2 and Tau protein (Figure 7).

To gain some more insight into the causes of differences between inhibitor-induced and FBP2 silencing-induced changes in the state of MT cytoskeleton, we checked the level of pTau also in HL-1 FBP2- cells and found a much smaller (5%) although statistically significant increment of pTau (*p* = 0.02, data not shown).

In turn, we found that under iFBP2 treatment the number of FBP2-MAP1B interactions in HL-1 cells was significantly increased (Figure 7) although the MAP1B protein level remained unchanged (data not shown).

Tubulin cytoskeleton is highly dynamic and undergoes quick cycles of assembly and disassembly. An imbalance in these processes may compromise the integrity of MT and lead to, among others, disturbances in mitochondria motility and distribution within the cell. The phosphorylation of Tau at Thr231 is critical for hyperphosphorylation of the protein [34] which in turn, reduces its affinity to MT and ability to polymerize them [30]. Furthermore, the MAP1B-induced MT stability is downregulated by phosphorylation [35].

The augmented cellular pool of pTau together with the increased number of Tau–FBP2 and also MAP1B–FBP2 interactions evoked by the iFBP2 treatment suggest that forced tetramerization of FBP2 induced perturbations in proteins involved in the MT–mitochondria interactions and ultimately led to deregulation of MT structure and mitochondria motility. In agreement with this suggestion, measurements of MT revealed that these structures were significantly shortened in iFBP2-treated HL-1 cells (Appendix A). However, although we observed a decrease in MT-mitochondria co-localization after the FBP2 tetramerization (Figure 8), only in the minority of iFBP2-treated cells did mitochondria tend to accumulate around the nucleus as shown in Figure 8 (iFBP2, left panel). In the majority of cells, mitochondria were shorter but quite evenly distributed along the shortened MT (iFBP2, right panel; see also in Figure 5).

In neurons, both the Thr231 residue of Tau and the MAP1B protein are primary targets of glycogen synthase kinase-3 beta (GSK3β) [34,35,36]. However, when we examined the effect of the short-term (1 h) treatment of cells with GSK3β inhibitor SB216763 on the motility of mitochondria in HL-1 cells, we found none (Figure 9A). Similarly, the 1h treatment of the cells with an inhibitor of phosphoinositide 3-kinase (PI3K)—an upstream effector (inhibitor) of GSK3—did not influence the rate of mitochondrial movement (Figure 9A).

Previously, we have shown that in HL-1 cells, inhibition of GSK3β dramatically increased FBP2-mitochondria interactions, which had a protective function on mitochondrial membrane potential [1]. Our present results suggest that this increase did not change mitochondrial motility. Perhaps the amount of FBP2 interacting with mitochondria in basal monoculture conditions was sufficient to support optimal mitochondrial dynamics.

### 3.6. Cardiac Fibroblasts-Derived Extracellular Vesicles Regulate Mitochondria-FBP2 Co-Localization and Mitochondrial Trafficking Rate

Results of our previous studies have demonstrated that during co-culture of cardiomyocytes and cardiac fibroblasts, orchestrated changes in expression and subcellular localization of glucose metabolism and lactate transport proteins could be observed in both cell types, supporting the existence of the fibroblasts-to-cardiomyocytes lactate shuttle. Moreover, as the HL-1 cells exhibit many features of the neonatal cardiac myocytes [12], the co-culture-induced changes in HL-1 cells resembled the postnatal switching from the “fetal” to “mature” cardiac phenotype. These changes were mediated by factors secreted to the culture medium in extracellular vesicles (EVs), and one of these changes was the shift of FBP2 from the nucleus to the cytoplasm in cardiomyocytes [10].

Here, we found that in HL-1 cardiomyocytes incubated with cardiac fibroblasts-derived EVs, the previously documented change of FBP2 localization from predominantly nuclear to cytoplasmic was correlated with an increase in co-localization of cytoplasmic FBP2-related and mitochondria-related fluorescent signals (mean M = 0.39, SD = 0.08 in control cells, vs M = 0.58, SD = 0.06 after the EVs treatment, *p* < 0.001; n = at least 130 cells in each condition). Consistently with our previous observations (Figure 3), these changes were accompanied by an increase in mitochondrial trafficking speed (Figure 9B).

In search of a factor that could be responsible for the observed changes, we analyzed the proteome of the EVs secreted by fibroblasts (Appendix A). Out of the identified proteins we selected 14-3-3 protein. 14-3-3 proteins are highly conserved, ubiquitous adaptor proteins that interact with serine/threonine-phosphorylated residues of a diverse array of signaling proteins and play important roles in a broad range of cellular processes. It is known that 14-3-3 proteins stabilize cAMP-dependent protein kinase A (PKA) as the holoenzyme (i.e., in its inactive state) [37], and we have previously shown that the inhibition of PKA resulted in the nucleo-cytoplasmic shuttling of FBP2 in HL-1 cells [13]. We found that the overnight incubation of HL-1 cells with PKA inhibitor had a similar accelerating effect on mitochondria mobility as the EVs treatment (Figure 9B). On the other hand, the incubation of HL-1 cells with EVs in the presence of adenylyl cyclase activator forskolin completely abolished their action (Figure 9B). This might suggest that in the process, the cAMP- and PKA-dependent pathway is involved.

The MT of HL-1 cardiomyocytes are less stable than MT from mature cells, and overstimulation of β-adrenergic receptors or the addition of forskolin can disrupt MT in HL-1 cells. Moreover, in neonatal cardiac myocytes, a stable pool of microtubules is involved in the beating activity, and stimulation of cAMP-dependent pathways might lead to the degradation of most labile MT [38]. The signals sent from fibroblast to cardiomyocytes in the co-culture might therefore stabilize the MT cytoskeleton, as part of the process of maturation of cardiomyocytes.

To test our assumption, we applied nocodazole on both untreated and iPKA-preincubated HL-1 cells and observed MT cytoskeleton. As could be expected [38], the mild nocodazole treatment perturbed the cytoskeleton. Preincubation with iPKA markedly reduced this perturbation (Figure 9C).

Together, it might be inferred that the EV-induced, PKA inhibition-dependent nucleus-to-cytoplasm/mitochondria shift of FBP2 and the associated changes in the mitochondrial trafficking rate and MT stability facilitate the maintenance of efficiently functioning mitochondria necessary for the rapid metabolism of increasing amounts of the fibroblasts-derived lactate and thus, protecting from pH drop that could result in reduction in cardiac myocytes contractility.

## 4. Conclusions

Mitochondrial network dynamics are important for the cell because of the crucial role of the organelles in ATP supply, calcium buffering and apoptosis regulation. The rearrangement of their network helps to meet the changing energy requirements of the cell. The proper trafficking and arrangement of mitochondria in cardiac (and other) cells depend on numerous interactions. Among them are the interactions of the outer mitochondrial membrane (OMM) with large protein complexes that facilitate long-distance trafficking of mitochondria along MT. The tubulin cytoskeleton interacts also with the voltage-dependent anion channel (VDAC) which regulates the permeability of the OMM and thus, mitochondrial metabolism (for review see [39]). However, the exact mechanism of this process is not fully elucidated. In contrast to neonate cardiomyocytes and cardiac cell lines, in mature cardiomyocytes, the motility of mitochondria is limited to the perinuclear pool of the organelles. Nevertheless, when MT undergo remodeling during adaptive and pathological changes in the myocardium, a perturbation in MT density and stability is sufficient to remodel the network of mitochondria [39]. Alterations in the network and mitochondrial trafficking also underlie a number of neuropathological diseases, including Alzheimer’s disease.

Previously, we have shown that in the cell culture, fibroblasts-to-cardiomyocytes cross-talk induced transitions that mimicked the postnatal shift from the “fetal” to “mature” mode of cardiac myocytes functioning. This included the translocation of FBP2 from the cardiomyocyte nucleus to cytoplasm where, as the dimer and/or the active R-state tetramer, it bound to mitochondrial proteins, including VDAC, and participated in the protection of the mitochondrial membrane [1,3,9,10,11,16].

Here we showed that chemically induced tetramerization of FBP2 resulting in reduction in FBP2-mitochondria interaction correlated with numerous changes in HL-1 cells: a decrease in mitochondrial membrane potential, increase in Tau Thr321 phosphorylation, augmentation of FBP2-Tau and FBP2-Map1B interactions, disturbance of tubulin network and tubulin–mitochondria interaction, marked reduction in mitochondrial membrane potential, velocity, and increase in mitophagy. In turn, induction of FBP2-mitochondria interactions by factors transported in EVs isolated from cardiac fibroblasts correlated with an increase in mitochondrial mobility, which could be mimicked by the inhibition of PKA but abolished by the activation of cAMP production. It therefore seems that the FBP2 molecules that interact with VDAC might be a part of a mechanism that influences cellular energetics by stabilizing mitochondria–MT interactions.

We are aware that this paper leaves the question of the precise, step-by-step mechanism of the observed phenomena unanswered. However, little is known about the roles of Tau and MAP1B and the mechanisms that regulate their function in cardiac cells. The information that FBP2 is a part of the microtubule stability-dependent trafficking of mitochondria in cardiomyocytes might therefore help to clarify these roles. Moreover, our findings highlighted the significance of the oligomeric state of FBP2 for the determination of the physiological roles of this multifunctional protein in the cell. Recognizing factors and mechanisms that modulate this state might be crucial for our understanding of processes that take place during physiological and pathological cardiac remodeling and during the onset of diseases that are rooted in the destabilization of MT and/or mitochondrial network dynamics.

## Figures and Tables

**Figure 1 cells-11-01710-f001:**
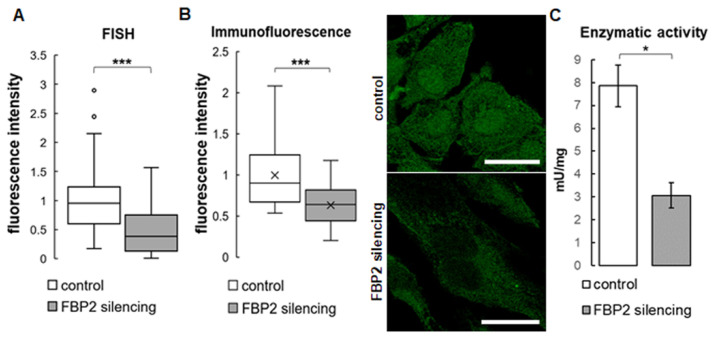
Results of a partial silencing of FBP2 expression in HL-1 cells. Quantification of fluorescence related to FBP2 mRNA level (**A**) and FBP2 protein (**B**), together with representative microscopic images of the protein distribution in the cells, and enzymatic activity measured in homogenates (**C**). Fluorescence was normalized to control group. Data are presented as median and interquartile range (**A**,**B**) or as mean and standard deviation (**C**). * *p* = 0.03; *** *p* < 0.001. Bar = 20 µm.

**Figure 2 cells-11-01710-f002:**
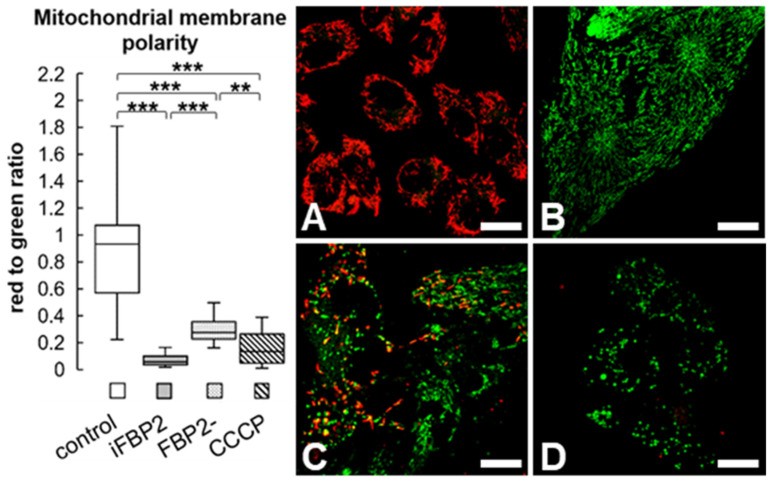
Mitochondrial membrane polarity in HL-1 cells. Representative microscopic images of cells stained with JC-1, and quantification of red to green fluorescence intensity in the cells. The fluorescence was normalized to control group. Data are presented as median and interquartile range. Control—untreated HL-1 WT cells (**A**), iFBP2—FBP2 tetramerizing agent-treated cells (**B**); FBP2-—cells with partially silenced expression of FBP2 (**C**), CCCP—HL-1 WT cells incubated with CCCP (**D**), a mitochondria depolarizing agent. ** *p* < 0.01 *** *p* < 0.001. Bar = 20 µm.

**Figure 3 cells-11-01710-f003:**
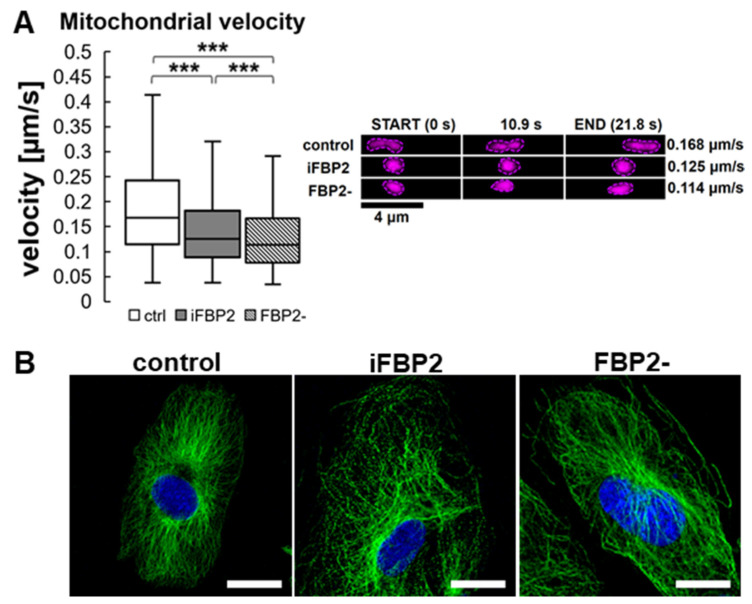
Mitochondrial velocity and microtubule cytoskeleton (MT) architecture in HL-1 cells. Quantification of mitochondrial velocity and graphical presentation of the velocity in a time range (**A**) and representative microscopic images of MT cytoskeleton (**B**) in different conditions. Ctrl—untreated HL-1 WT cells, iFBP2—FBP2 tetramerizing agent-treated cells; FBP2-—cells with partially silenced expression of FBP2. Data are presented as median and interquartile range. *** *p* < 0.001. Bar = 20 µm.

**Figure 4 cells-11-01710-f004:**
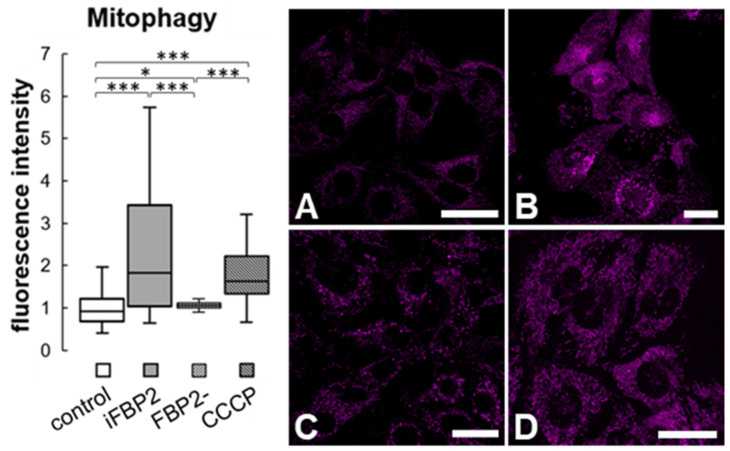
Mitophagy in HL-1 cells. Quantification of fluorescence intensity related to mitophagy and representative images of cells stained using Mitophagy Detection Kit. The fluorescence was normalized to control group. Control—untreated HL-1 WT cells (**A**), iFBP2—FBP2 tetramerizing agent-treated cells (**B**); FBP2-—cells with partially silenced expression of FBP2 (**C**), CCCP—HL-1 WT cells incubated with CCCP, a mitophagy inducing agent (**D**). Data are presented as median and interquartile range. * *p* < 0.05 *** *p* < 0.001. Bar = 20 µm.

**Figure 5 cells-11-01710-f005:**
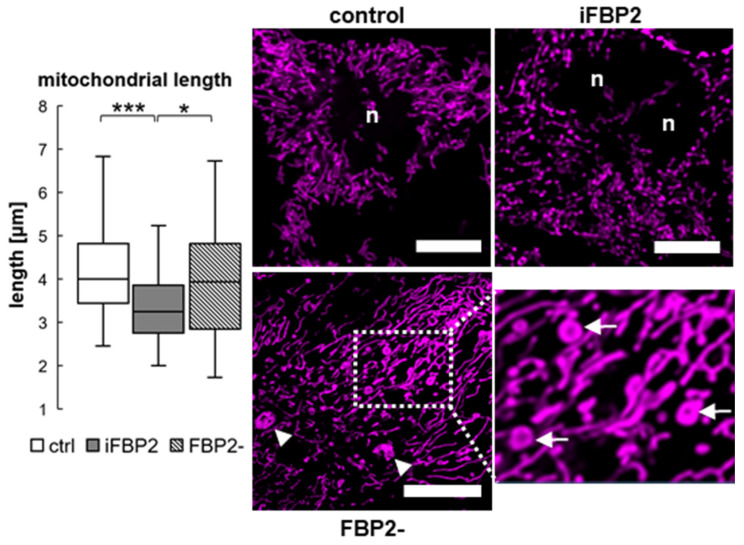
Mitochondrial length and morphology in HL-1 cells. Quantification of length of the organelles and representative microscopic images presenting the shape of mitochondrial network in different conditions. Ctrl—untreated HL-1 WT cells, iFBP2—FBP2 tetramerizing agent-treated cells; FBP2-—cells with partially silenced expression of FBP2; n—nucleus. Arrows point to donut-shaped mitochondria (in the magnified part of the FBP2- image), arrowheads point to aggregates observed in HL-1 FBP2- cells. Data are presented as median and interquartile range. * *p* < 0.05 *** *p* < 0.001. Bar = 10 µm.

**Figure 6 cells-11-01710-f006:**
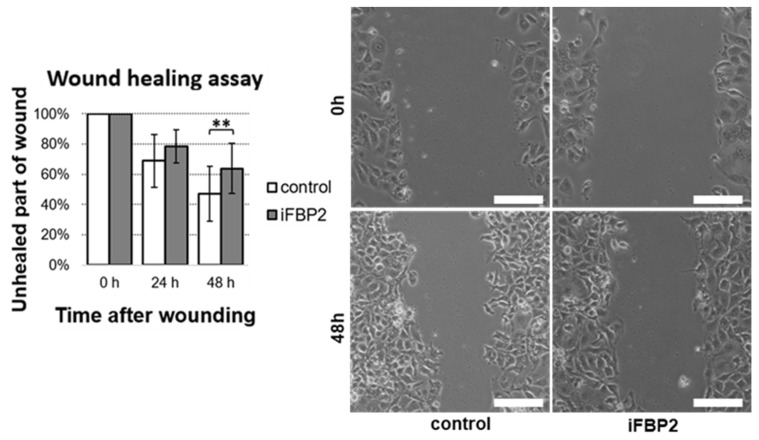
Wound healing assay (scratch test). Quantification of measurements of unhealed part of wounds from three time points and representative images of untreated HL-1 (control) and the FBP2 tetramerizing agent-treated cells (iFBP2) taken immediately after wounding and after 48 h of cells growth. Data are presented as mean and standard deviation. ** *p* < 0.01. Bar = 100 µm.

**Figure 7 cells-11-01710-f007:**
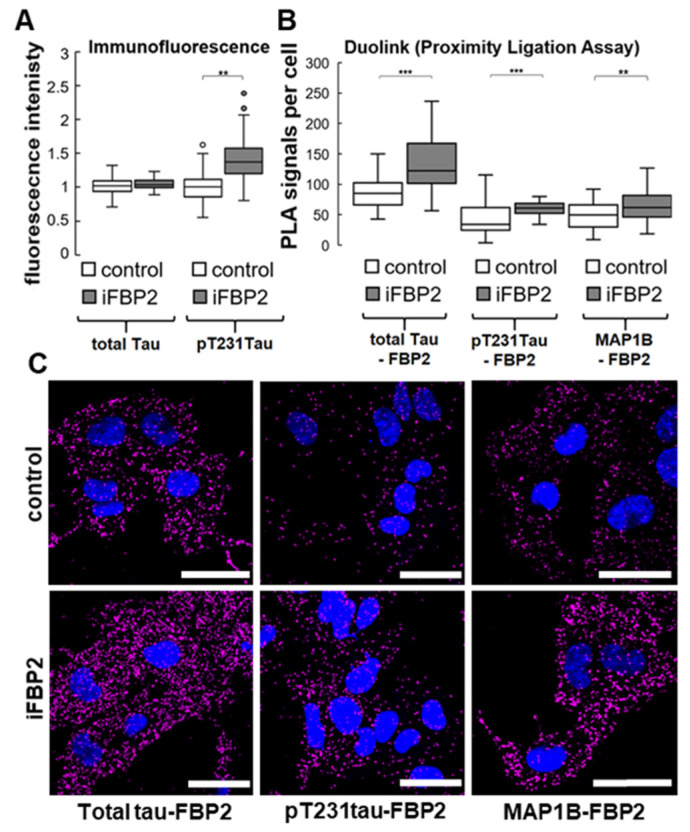
Increase in pT321 Tau as well as Tau- and MAP1B–FBP2 interactions in HL-1 cells after tetramerization of FBP2. (**A**) Quantification of immunofluorescent signals related to T321-phosphorylated and total Tau. Fluorescence was normalized in control group. (**B**) Tau– and MAP1B–FBP2 interactions measured by the Proximity Ligation Assay together with (**C**) representative confocal images, where magenta signal represents sites of the interactions, and nuclei are in blue. Control—untreated HL-1 cells, iFBP—FBP2 tetramerizing agent-treated cells. Data are presented as median and interquartile range. Dots represent outliers amongst data. ** *p* < 0.01 *** *p* < 0.001. Bar = 30 µm.

**Figure 8 cells-11-01710-f008:**
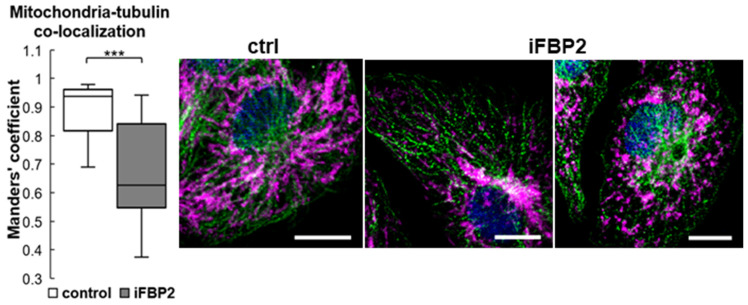
Tetramerization of FBP2 correlates with a decrease in mitochondria-tubulin co-localization. Control (ctrl)—untreated HL-1 cells; iFBP2—FBP2 tetramerizing agent-treated cells. In zoomed microscopic images, mitochondria are presented in magenta, tubulin in green and the nucleus in blue. Measurements were taken from 12 randomly selected areas from each microscopic slide. Data are presented as median and interquartile range. *** *p* < 0.001. Bar = 10 µm.

**Figure 9 cells-11-01710-f009:**
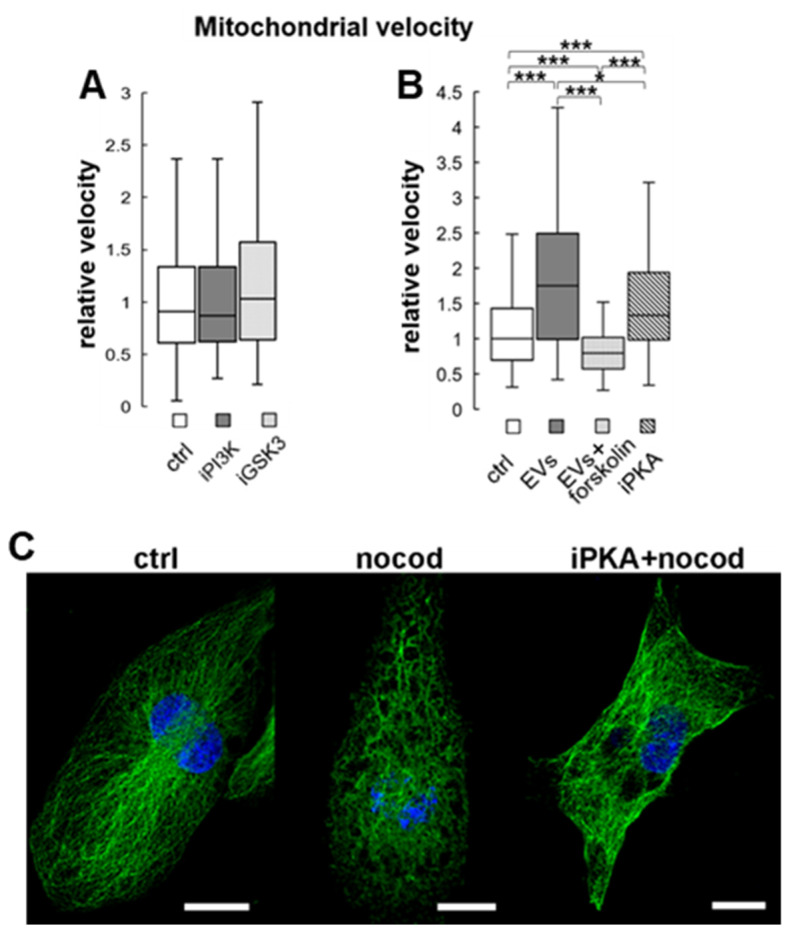
Changes in mitochondrial velocity under different treatments, and effects of nocodazole and PKA inhibitor on tubulin cytoskeleton in HL-1 cardiomyocytes. (**A**) The changes observed after 1h of PI3K or GSK3 inhibitor treatment. (**B**) The changes induced by extracellular vesicles (EVs) from cardiac fibroblasts and PKA pathway effectors. Data were normalized to control, and are presented as median and interquartile range. * *p* < 0.05, *** *p* < 0.001. (**C**) The effect of PKA inhibitor on nocodazole-induced disruption of MT cytoskeleton. Bar = 10 µm. Ctrl—untreated cells; EVs—extracellular vesicles-treated cells; EVs + forskolin—cells preincubated with adenylyl cyclase activator forskolin and treated with EVs; iPKA—cells treated with an inhibitor of the cAMP-dependent protein kinase A, nocod—cells treated with nocodazole; iPKA+nocod—cells preincubated with iPKA before the nocodazole treatment.

## Data Availability

Not applicable.

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
