# Peer review of "FBP2—A New Player in Regulation of Motility of Mitochondria and Stability of Microtubules in Cardiomyocytes"

_cells, 2022, doi:10.3390/cells11101710_

Round 1

Reviewer 1 Report

Overview

Pietras et al. have investigated the multi-functions of the metabolic muscle enzyme fructose 1.6 biphosphatase 2 (FBP2) in HL-1 cardiomyocytes. Besides its main enzymatic reaction in energy metabolism of conversion of FBP to G3P + DHAP, FBP2 also regulates tau phosphorylation, interactions between FBP2/tau and FBP2/MAP1B, affects polymerization of microtubules, mitochondrial membrane potential, intracellular mitochondrial transport, and prevents mitophagy. Removal of about 40% of FBP2 in HL-1 cells by either shRNA silencing or use of an inhibitor of FBP2 (iFBP2) removes these protective multifunctions of the enzyme, leading to disorganization of the microtubule and mitochondrial networks. The shRNA silencing does not alter the dimer/tetramer ratio of FBP2 in the cell, but the iFBP2 inhibition converts FBP2 into its inactive tetramer form. The Gizak lab has been very active in investigating FBP2 over many years and has an excellent publication record in good journals. The present manuscript is well-written, and the experiments are well thought out with proper controls and thorough statistical analyses of the results.

Major comments

  1. As the two different downregulation techniques, shRNA silencing and iFBP2 inhibition, have different mechanisms, one preserving the dimer structure of FBP2 and the other leading to tetramerization of FBP2, a bit more information on the known effects of dimer to tetramer conversion could be added to the Introduction with a relevant reference cited. For example, in muscle, T(+AMP) which is inactive changes to T(-AMP), then to the R active form of the tetramer. This involves a major twist in the upper dimer of the tetramer? Does this process also occur in HL-1 cardiomyocytes? Have the dimer/tetramer ratios been measured in cardiomyocytes? What is the mechanism of tetramer entry into the nucleus, which doesn’t occur with the dimer?
  2. Given that the shRNA silencing of FBP2 decreased mitochondrial polarity and motility, but had no or little effect on mitophagy, does this suggest that the important driver of mitophagy is the shortened length of mitochondria, as seen in the iFBP2-treated cells? Both forms of down-regulation of FBP2 led to appearance of donut-shaped mitochondria, and one might think that this would lead to mitophagy, although the mitochondria had good length after silencing of 36% of the FBP2.

3     Does iFBP2 have any off-target effects that might contribute to the differences between shRNA knockdown and enzyme inhibition?

  1. Does the increase in phosphorylated tau have any measurable negative effects on the cardiomyocyte? Would the increased tau phosphorylation and MAP-1B-FBP2 interactions after iFBP2 treatment be expected to be the cause of the altered microtubule network? Phosphorylated MAP-1B is believed to contribute to microtubule bundling in the growth cone of neurons and stabilize microtubules as well as link them together. Whereas normal Tau stabilizes microtubules, pTau destabilizes them.
  2. On line 487, please define the action of the 14-3-3 proteins such as “The 14-3-3 proteins are adapter proteins that interact with phospho-serine and threonine motifs in diverse binding proteins.”

Minor comments, typographical errors, and style change suggestions

  1. The word “Data” is plural. Throughout the manuscript, mostly in the figure legends, change “Data is” or “was” to “data are” or “were”.

2     Replace “what” with “which” in Lines 47, 54, 225, 255, 361, 389, 430, 452, 520, and 542.

Line 9       Delete “Quite”

Line 72     TMS-016), 2 mM

Line 156   mitochondria were analyzed using

Line 185   Data were

Line 191   Data are

Line 196   FBP@-interacting proteins, affinity

Line 208   ER cisterns) and its local

Line 218   exception was toxic

Line 224   mitochondria, the organelles

Line 251   polarity [1], we tested

Line 295   A 1-h incubation

Line 327   shape/length to that of HL-1

Line 472   condition).

Line 475   different treatments, and

Line 514   regulation. Rearrangement

Line 534   mitochondrial membrane [1, ...

Line 536-7   HL-1 cells: increase in Tau…

In Supplementary Materials, Fig. S3 legend      in each condition. *** p

Author Response

We found the comments and suggestions very helpful, and we changed the manuscript accordingly. Our detailed responses along with a list of changes in the text are below.

Reviewer 1.

Major comments

  1. As the two different downregulation techniques, shRNA silencing and iFBP2 inhibition, have different mechanisms, one preserving the dimer structure of FBP2 and the other leading to tetramerization of FBP2, a bit more information on the known effects of dimer to tetramer conversion could be added to the Introduction with a relevant reference cited. For example, in muscle, T(+AMP) which is inactive changes to T(-AMP), then to the R active form of the tetramer. This involves a major twist in the upper dimer of the tetramer. Does this process also occur in HL-1 cardiomyocytes? Have the dimer/tetramer ratios been measured in cardiomyocytes? What is the mechanism of tetramer entry into the nucleus, which doesn’t occur with the dimer?

Response: HL-1 cardiomyocytes express only muscle FBP (PMID: 19250949) and thus, one might expect that the twist also occurs in the protein there, since it is typical to this isozyme. Importantly, in this “twisted” R-state of FBP2, additional surfaces are exposed to the solution and hence, they may form new docking sites for binding partners which may explain the diversity of FBP2 interactions/functions in the cell. Since both dimer and tetramer can exist in cytoplasm and nucleus, it is hard to measure their ratios in the cell.

Studies with mutant FBP2 that cannot tetramerize have revealed that only the dimeric form appears to interact with mitochondria but both dimers and tetramers are imported to the nucleus (the Nuclear Localization Sequence is present within each FBP2 monomer). In turn, the putative nuclear export signal is located on the surface of interaction between the upper and the lower dimers within the tetramer and thus, dimers are readily exported to the cytoplasm (PMID: 29383170). It cannot be excluded, however, that dimers bound to some nuclear components might be retained in the compartment, and that tetramers are exported using other mechanisms (e.g., the piggy-back mechanism).

According to the Reviewer’s suggestion, we added to the Introduction (page 1-2):

„In mammalian tissues, two different FBP isozymes are expressed: the liver FBP (FBP1) and muscle (FBP2) isozyme. They catalyze hydrolysis of fructose-1,6-phosphate to fructose-6-phosphate and inorganic phosphate. Both isozymes form homotetramers. In the presence of their allosteric inhibitors – AMP and NAD+, FBP1 and FBP2 tetramers adopt a similar, inactive T-state in which two upper subunits (the upper dimer) are slightly rotated in respect to the lower two subunits (the lower dimer). In the absence of the inhibitors, tetramers of both isozymes adopt the active R-state. But while the FBP1 tetramer is almost flat (for review see [7]), FBP2 adopts a unique cross-like structure in which the upper dimer is twisted about 90o in respect to the lower one [8]. FBP2 may also exist as the dimer which is fully active and is not inhibited by AMP and NAD+ since the mechanism of the allosteric inhibition requires the presence of the tetrameric conformation: binding of AMP to subunits within one dimer inhibits the catalytic sites in the second dimer. Importantly, both in the unique R-state of FBP2 and in its dimeric form, additional surfaces are exposed to the solution and hence, they may form new (as compared to FBP1 and also to the T-state of FBP2 tetramer) docking sites for binding partners (for review see [7]). “

We also added two citations to the reference list and changed the numbering accordingly.

As a consequence of these changes, we modified the next sentence:

“In HL-1 cardiomyocytes and KLN205 squamous cell carcinoma cell lines, partial silencing of the muscle isozyme of FBP (so-called FBP2, as opposed to the liver isozyme – FBP1) expression leads to sensitization of mitochondria to membrane-depolarizing stress conditions [1,3].”

  1. Given that the shRNA silencing of FBP2 decreased mitochondrial polarity and motility, but had no or little effect on mitophagy, does this suggest that the important driver of mitophagy is the shortened length of mitochondria, as seen in the iFBP2-treated cells? Both forms of down-regulation of FBP2 led to appearance of donut-shaped mitochondria, and one might think that this would lead to mitophagy, although the mitochondria had good length after silencing of 36% of the FBP2.

Response: In addition to shortening of mitochondria, forced tetramerization of FBP2 by the inhibitor correlated also with the significant disturbance of microtubule cytoskeleton, so it is likely that only the cumulative effect of all changes induced by iFBP2 was sufficient to induce mitophagy. However, we cannot exclude that the tetramerization of FBP2 resulted in additional changes leading to increased autophagy of mitochondria.

Although mitochondria in the iFBP2-treated cells were short and sometimes spherical, the donut-shaped mitochondria were frequent only in FBP2-silenced cells. Donut-shaped mitochondria are immobile, but they have higher tolerance to an increase in matrix volume than the “linear” organelles, they can also recover faster, when cellular stress disappears (PMID: 21372848), therefore they are more mitophagy-resistant.

3     Does iFBP2 have any off-target effects that might contribute to the differences between shRNA knockdown and enzyme inhibition?

Response: The iFBP2 is an commercially available inhibitor that was tested using numerous targets, but of course, one can never completely rule out inhibition of other targets. Taking into account the “additional” (compared to shRNA) effect of iFBP2-treatment on MT structure one might wonder if the inhibitor can directly disturb MT structure, but several differences in the structure of iFBP2 and nocodazole – a popular MT-destabilizing agent (e.g., the presence of two benzene rings in iFBP2, and only one in nocodazole) suggest that this is not the case.

  1. Does the increase in phosphorylated tau have any measurable negative effects on the cardiomyocyte? Would the increased tau phosphorylation and MAP-1B-FBP2 interactions after iFBP2 treatment be expected to be the cause of the altered microtubule network? Phosphorylated MAP-1B is believed to contribute to microtubule bundling in the growth cone of neurons and stabilize microtubules as well as link them together. Whereas normal Tau stabilizes microtubules, pTau destabilizes them.

Physiological roles of Tau and (especially) MAP1B in heart are poorly understood. Presence of the proteins in cardiac cells has been confirmed quite recently (in the last 7 years) and studies on their role in the heart are scarce.

In one paper (PMID: 28059795), Tau deficient mice showed cardiac hypertrophy, reduced left atrial contractility and chronically increased blood pressure, what resembled an early aging phenotype. The authors did not study mechanisms of observed changes, but pointed that MTs play an important role in development of cardiac hypertrophy (their assembly is attenuated).

In turn, Tsai et al. (PMID: 33810615) observed changes in MAP1B expression in patients with abnormality of mitral valve and the left ventricle failure.

However, regardless of the lack of experimental data one might expect that both proteins play the MT stability-regulating role also in cardiac cells.

Phosphorylated Tau is soluble and cannot stabilize MTs. Although phosphorylation of MAP1B might not change its ability to bind to MTs, it is needed to maintain proper dynamics of MTs in neurons (for review see: PMID: 28980356). It has been also suggested that in non-neuronal cells, phospho-MAP1B might increase the population of unstable microtubules (see in PMID: 15731007).

Thus, as we see it, FBP2 tetramerization leads to its increased interaction with MAP1B and Tau what probably reduce their “normal” interactions with MTs and influence interactions with other proteins (e.g., kinases), contributing to deregulation of MT dynamics. On the other hand, tetramerized FBP2 cannot bind to mitochondrial proteins – e.g., VDAC which also interacts with MTs to regulate permeability of outer mitochondrial membrane – and together, destabilization of all these interactions leads to observed changes in MTs structure and mitochondrial motility.

  1. On line 487, please define the action of the 14-3-3 proteins such as “The 14-3-3 proteins are adapter proteins that interact with phospho-serine and threonine motifs in diverse binding proteins.”

Response: We added the required information to the text and we modified original sentences to match introduced changes. Now the text reads:

“Out of the identified proteins we selected 14-3-3 protein. 14-3-3 proteins are highly conserved, ubiquitous adaptor proteins that interact with serine/threonine-phosphorylated residues of a diverse array of signaling proteins and play important roles in a broad range of cellular processes. It is known that 14-3-3 proteins stabilize cAMP-dependent protein kinase A (PKA) as the holoenzyme (i.e. in its inactive state) [37], and we have previously shown that inhibition of PKA resulted in the nucleo-cytoplasmic shuttling of FBP2 in HL-1 cells [13].”

Minor comments, typographical errors, and style change suggestions

  1. The word “Data” is plural. Throughout the manuscript, mostly in the figure legends, change “Data is” or “was” to “data are” or “were”.

2     Replace “what” with “which” in Lines 47, 54, 225, 255, 361, 389, 430, 452, 520, and 542.

Line 9       Delete “Quite”

Line 72     TMS-016), 2 mM

Line 156   mitochondria were analyzed using

Line 185   Data were

Line 191   Data are

Line 196   FBP@-interacting proteins, affinity

Line 208   ER cisterns) and its local

Line 218   exception was toxic

Line 224   mitochondria, the organelles

Line 251   polarity [1], we tested

Line 295   A 1-h incubation

Line 327   shape/length to that of HL-1

Line 472   condition).

Line 475   different treatments, and

Line 514   regulation. Rearrangement

Line 534   mitochondrial membrane [1, ...

Line 536-7   HL-1 cells: increase in Tau…

In Supplementary Materials, Fig. S3 legend      in each condition. *** p

We corrected the above-listed errors.

Reviewer 2 Report

For the reasons described below I do not recommend publication of the submitted manuscript at its current state. The authors need to reevaluate the presentation of the study and provide more data.

Major Points

From the text in lines 52-56 in the introduction, the reader understands that the mitochondria clustering is due to the expression of the FBP2 variant Val115Met associated with remitting leukodystrophy. However, in the next paragraph, lines 57-64, the authors treat the mitochondria clustering as a normal behaviour that is dependent on the presence of normal levels of dimeric as opposed to the tetrameric FBP2. What is the connection between the  Val115Met variant and the dimeric vs tetrameric state of FBP2? Is the Val115Met a tetramer and/or is it capable of forming tetramers?

The way the manuscript is written creates a confusion to the reader as to what are the effects of the depletion of the FBP2  protein and what are the results with the induction of FBP2 tetramerization by iFBP2 and whether the results can be compared with each other.

Based on the summary of the observations shown below it becomes obvious that the results can not be considered comparable because  either data are missing or they are contradictory.

Studied

FBP2 Depletion

iFBP2

 levels in cytoplasm

Yes reduction

????

nuclear staining/tetramrer

Yes reduction

????

 enzymatic activity

Yes reduction

????

membrane polarity

Yes affected

Yes affected

 velocity

25% reduction

25% reduction

mitophagy

No

Yes

Mitochondria Length

No; donut shaped

Yes

Wound Healing

????

Delay

MT network destabilisation

No !!!

Yes !!!

Soluble Tau phospho

????

27% increase

Tau or MAP1B - FBP2 colocal

????

Increased

Mitoch_MT colocalization

????

Lost

The only two common observation after depletion FBP2 or FBP2 inhibition by iFBP2 are the loss of Mitochondrial membrane polarity and velocity. So one can conclude that mitophagy is not dependent on velocity reduction (!! ??).  The rest of the observed phenotypes either they were not examined by the authors or they were not the same. So, it is confusing to the reader why the authors present their data in that manner and why the do not focus only in one of the conditions.

To this reviewer, it is curious that the authors preferred to evaluate RNAi depletion of  FBP2 by FISH and immunofluorescence microscopy especially since it seems the straining is diffused cytoplasmic and nuclear. Why standard Western blotting or PCR techniques were not used?.  Is there a reason? The reported differences between depleted vs inhibited FBP2 raise doubts about  the efficacy of RNAi depletion or the target specificity of iFBP2 (FBP2 vs MT/tubulin).

Does the control in Fig 1 refer to control infection or to untreated cells? 40 % reduction by FISH and 36% reduction in protein levels by IF are enough to draw safe conclusions about FBP2 functions  due to its depletion. Only the 25% reduction of mitochondria velocity and the membrane polarisation were affected under the FBP2 depletion conditions. Can it be that the observed effects are the result of stress conditions due to shRNA infection conditions and not due to depletion of FBP2?

One aspect that the authors do not consider is that the observed MT cytoskeleton changes may not be related directly to mitochondria derived effects  due to FBP2 loss of function but by a direct effect of iFBP2 inhibitor  on microtubules per se, independent of FBP2 tetramerization. The low nocodazole phenotype resembles that of iFBP2 phenotype (Fig 9).

In the paragraph lines 285-287 the authors present the results showing that there is a 25% reduction  in mitochondria motility (Fig 3) after depletion of FBP2 and treatment with  iFBP2, and at the same time show that the levels of P-tau (and therefore soluble tau) increase which means that there less tau on Mts. However, previous studies have shown that the presence of tau inhibit mitochondrial trafficking on Mts dependent by MT-based motor proteins (J Cell Biol. 1998 Nov 2;143(3):777-94. doi: 10.1083/jcb.143.3.777).  Therefore loss of MT associated tau should enhance motility.. Similarly faster mitochondrial retrograde transport was found in developing MAP1B−/− neurons (Biochem. J. (2006) 397, 53–59 doi:10.1042/BJ20060205).  The authors should reconcile the two opposite observations

The data with iFBP2 show an interphase microtubule network destabilisation effect and the presence of a subpopulation of shorter MT cytoskeleton similar to the low nocodazole resistant MTs as shown in Fig9C. The low nocodazole resistant MTs are known to be rich in post translationally modified alpha tubulin,  such as detyrosinated and acetylated alpha tubulin. The authors need to test if indeed is the case fo the iFBP2  treated cells. There is the possibility that the MT induced destabilization in the presence of iFBP2 may not be directly related to  inhibition of FBP2, especially since as depletion of FBP2 does not produce the same MT destabilization.

Deacetylation of α-tubulin reduces the efficiency with which motor proteins bind to microtubules and this could lead to mitochondria transport more or less similar to the defects observed by the authors (doi.org/10.1016/j.cub.2006.09.014). At the same time though, data strongly indicate that HDAC6 plays an important role in mitochondrial transport through its deacetylation effect on Miro1, a well-known partner in the Miro/Milton complex that links mitochondria to motor proteins. Deacetylation of Miro1 has been previously shown to decrease mitochondrial transport in axons (J Cell Biol 019 Jun 3;218(6):1871-1890 doi: 10.1083/jcb.201702187). Therefore, HDAC6 inhibition would increase the acetylated Miro1 and may restore the observed loss of motility of mitochondria following depletion or inhibition of FBP2. The authors may test this possibility.

Since  iFBP2 induces tetramerization of FBP2 then where does the LPA tetramer-Phospo-tau complex colaclize? (Not on mitochondria not on MTs)

Given the data presented in Fig 8 and the decrease in MT-mitochondria co-localization after FBP2 tetramerization, is in’t surprising that there is only 25% decrease in the velocity of Mitochondria displacement on Mts?

The presented data in figure 7 argue for enhanced interaction of tetrameric FBP2 with tau protein which also argues for the presence of tetrameric FBP2 in the cytoplasm. Why the iFBP2 induced FBP2 tetramers are not nuclear? Does FBP2 in iFBP2 treated cells all become nuclear since it become tetrameric?

Where are the data supporting the statement in lines 298-301. The dimer (cytoplasmic) to tetrameric (nuclear) in untreated cells can be determined but the dimer to tetramer ratio  in cytoplasm it is impossible to determine by the methods presented here.

Minor points

The manuscript needs to be edited by a native English speaking editor since there is a lot of English language editing is required. Below I pinpoint some of the necessary editions.

Line 36 replace ‘such’ to ‘certain’

Line 47 replace ‘what’ to ‘and’

Line 53 edit sentence to  ‘ fibroblasts carrying the variant Val115Met, …'

Line 54 replace ‘what’ to ‘and’

Line 55 edit sentence to “In the majority of the the cells….”

Line 218 edit “The only exception were hypoxic…’ to “The only exception was hypoxic…’

Line 225 replace ‘what’ to ‘and’

Figure 1, Panel B ,Y axis : correct   title to ‘intensity’

From the material and Methods section as well from the result section and figure legend it is not stated after how many hours or days following RNAi depletion the data are presented.

Line 273 edit ‘Quite a some time ago, …’ to ‘We have previously reported…’

Line 293 edit ‘Data is’ to ‘Data are’

Line 327 edit ‘that’ to ‘compared to…’

Line 361 replace ‘what’ to ‘and’

Edit sentences 429-430 to ‘….induced perturbations in proteins involved in the MT–mitochondria interactions and  ultimately led to changes in the MT cytoskeleton and mitochondria motility.’

Line 447 Correct  ‘Figure 8A’ to ‘Figure 9A’

Line 452 replace ‘what’ to ‘and’

Author Response

Reviewer 2

We would like to thank the Reviewer for interesting and exhaustive discussion, and for the necessary criticism that undoubtedly helped us to improve the quality of the manuscript. However, we must admit that we found some of the arguments raised in the review confusing.

We are not sure why the Reviewer compared the effect of FBP2 silencing to its inhibition (the table presented by the Reviewer). Our manuscript is not a systematic study on the differences in the effects of FBP2 tetramerization and silencing but the mechanistic study on the new role of FBP2; about the enzyme role in motility of mitochondria and stability of microtubules in cardiomyocytes.

  1.  “From the text in lines 52-56 in the introduction, the reader understands that the mitochondria clustering is due to the expression of the FBP2 variant Val115Met associated with remitting leukodystrophy. However, in the next paragraph, lines 57-64, the authors treat the mitochondria clustering as a normal behaviour that is dependent on the presence of normal levels of dimeric as opposed to the tetrameric FBP2.

What is the connection between the  Val115Met variant and the dimeric vs tetrameric state of FBP2? Is the Val115Met a tetramer and/or is it capable of forming tetramers?”

Response: We did not suggest anywhere in the text that mitochondria clustering is “a normal behaviour which depends on the presence of dimeric FBP2”. We simply tried to explain which of our earlier observations prompted us to investigate the involvement of FBP2 in mitochondrial motility in the cell:

“Therefore, we sought to clarify the contribution of FBP2 to this clustering of mito- 57
chondria in HL-1 cardiomyocytes – the cells that express only the FBP2 isozyme, and in 58
which the pathways regulating intracellular behavior of the protein was best studied. We 59
found that chemically induced tetramerization of FBP2 resulting in reduction of FBP2- 60
mitochondria interaction correlated with a disturbance of tubulin network, marked re- 61
duction of mitochondrial mobility and increase in mitophagy. In turn, induction of FBP2- 62
mitochondria interactions by EVs isolated from cardiac fibroblasts correlated with an in- 63
crease of mitochondrial mobility.”

And as we described in the paper mentioned by the Reviewer, Val115Met variant of FBP2 can adopt tetrameric structure, however, it is less susceptible to the allosteric inhibition and less thermodynamically stable than the WT FBP2.

  1. “The way the manuscript is written creates a confusion to the reader as to what are the effects of the depletion of the FBP2  protein and what are the results with the induction of FBP2 tetramerization by iFBP2 and whether the results can be compared with each other.

Response: As we mentioned above, the manuscript is not about the comparison of the effects of FBP2 silencing and inhibition/tetramerization. We explained that although both partial silencing and tetramerization of FBP2 resulted in reduction of mitochondrial velocity, the silencing - that reduced the overall amount of FBP2 protein but apparently, did not change the dimer-tetramer ratio in the cytoplasm – did not alter significantly mitochondrial length, mitophagy and the MT cytoskeleton architecture. Therefore, we focused on the effects of the FBP2 inhibitor/tetramerizing factor – the change in dimer/tetramer ratio in the cell.

To make our intentions clearer, we added the sentence to the revised version of the article (page 11): “Therefore, in subsequent experiments, we focused on the effects of changing the dimer-tetramer ratio in the cell.”

  1. “Based on the summary of the observations shown below it becomes obvious that the results can not be considered comparable because  either data are missing or they are contradictory.

Studied

FBP2 Depletion

iFBP2

 levels in cytoplasm

Yes reduction

????

nuclear staining/tetramrer

Yes reduction

????

 enzymatic activity

Yes reduction

????

membrane polarity

Yes affected

Yes affected

 velocity

25% reduction

25% reduction

mitophagy

No

Yes

Mitochondria Length

No; donut shaped

Yes

Wound Healing

????

Delay

MT network destabilisation

No !!!

Yes !!!

Soluble Tau phospho

????

27% increase

Tau or MAP1B - FBP2 colocal

????

Increased

Mitoch_MT colocalization

????

Lost

Response: Again, there are no “missing and contradictory” data, since our work was not intended to be a systematic study comparing the effects of FBP2 tetramerization and silencing.

We did not demonstrate the mentioned effects of FBP2 inhibition  (the first 3 positions in the Reviewer’s table) and the explanation is very simple: there is no need to show that FBP2 inhibitor in fact inhibits FBP2 activity - it is a commercially available inhibitor and its proprieties and effects on FBP2 nucleo-cytoplasmic localization has been demonstrated in several papers, also published by our group.

We observed differences (no effect of the silencing but evident changes after the inhibition/tetramerization) in mitophagy, mitochondria length and MT network destabilization and this is why we tested the effects of the inhibitor on such parameters as: Tau phosphorylation, Tau or MAP1B-FBP2 colocalization, mitochondria-MT colocalization and wound healing. What could be the rationale for studying these parameters if there were no alterations in the MT network and mitochondrial length?

  1. “The only two common observation after depletion FBP2 or FBP2 inhibition by iFBP2 are the loss of Mitochondrial membrane polarity and velocity. So one can conclude that mitophagy is not dependent on velocity reduction (!! ??).  The rest of the observed phenotypes either they were not examined by the authors or they were not the same. So, it is confusing to the reader why the authors present their data in that manner and why the do not focus only in one of the conditions.”

Response: As we pointed above, the effects of the FBP2 silencing and use of the inhibitor (that tetramerizes the enzyme in its inactive T-state) are simply different. One can conclude that the cumulative changes induced by forced tetramerization of FBP2 were sufficient to significantly increase mitophagy but changes induced just by reduction of FBP2 cellular amount (not the dimer-tetramer ratio) were not.

  1. To this reviewer, it is curious that the authors preferred to evaluate RNAi depletion of  FBP2 by FISH and immunofluorescence microscopy especially since it seems the straining is diffused cytoplasmic and nuclear. Why standard Western blotting or PCR techniques were not used?.  Is there a reason?”

Response: In our experiments, the most important thing was to check the reduction in the amount of FBP2 protein in the cell. This was achieved by quantification of FBP2-related immunofluorescent signal and measurement of the enzymatic activity. But additionally, we tested the level of FBP2 mRNA using FISH and obtained results were consistent. Both FISH and IF represent the same level of the “uncertainty” of the quantitation as PCR and WB techniques. The enzyme activity measurement is obviously the most accurate assay.

The reported differences between depleted vs inhibited FBP2 raise doubts about  the efficacy of RNAi depletion or the target specificity of iFBP2 (FBP2 vs MT/tubulin).”

Response: Taking into account that after the silencing, the enzyme activity demonstrated even deeper drop in the amount of FBP2 than FISH and IF, we cannot see any reasons to doubt in the efficacy of silencing. And our feeling is that the Reviewer share our lack of doubts, because the next paragraph of the review reads: “40 % reduction by FISH and 36% reduction in protein levels by IF are enough to draw safe conclusions about FBP2 functions  due to its depletion”. And, once more, we would like to stress that the observed differences between iFBP2 and shRNA treatment are the differences between the forced tetramerization of FBP2 molecules and reduction of the total amount of FBP2 protein.

  1. “Does the control in Fig 1 refer to control infection or to untreated cells? 40 % reduction by FISH and 36% reduction in protein levels by IF are enough to draw safe conclusions about FBP2 functions  due to its depletion. Only the 25% reduction of mitochondria velocity and the membrane polarisation were affected under the FBP2 depletion conditions. Can it be that the observed effects are the result of stress conditions due to shRNA infection conditions and not due to depletion of FBP2?”

Response: We do not expect that the changes are simply linear although of course, they are proportional. As the procedure of stably transfected cells selection is long-lasting thus, it does not seem that the observed changes resulted from the transfection-related stress conditions. Additionally, transfection with scrambled shRNA did not impact negatively HL-1 cells.

  1. One aspect that the authors do not consider is that the observed MT cytoskeleton changes may not be related directly to mitochondria derived effects  due to FBP2 loss of function but by a direct effect of iFBP2 inhibitor  on microtubules per se, independent of FBP2 tetramerization. The low nocodazole phenotype resembles that of iFBP2 phenotype (Fig 9).

Response: Of course, it cannot be completely ruled out that an inhibitor, even if commercially available and tested using numerous targets, will not inhibit other targets. Several differences in the structure of the two inhibitors (e.g., the presence of two benzene rings in iFBP2, and only one in nocodazole) suggests that this is not the case.

  1. “In the paragraph lines 285-287 the authors present the results showing that there is a 25% reduction  in mitochondria motility (Fig 3) after depletion of FBP2 and treatment with  iFBP2, and at the same time show that the levels of P-tau (and therefore soluble tau) increase which means that there less tau on Mts. However, previous studies have shown that the presence of tau inhibit mitochondrial trafficking on Mts dependent by MT-based motor proteins (J Cell Biol. 1998 Nov 2;143(3):777-94. doi: 10.1083/jcb.143.3.777).  Therefore loss of MT associated tau should enhance motility.. Similarly faster mitochondrial retrograde transport was found in developing MAP1B−/− neurons (Biochem. J. (2006) 397, 53–59 doi:10.1042/BJ20060205).  The authors should reconcile the two opposite observations”

Response: We assume that the Reviewer agrees with the scientific literature showing that the role of Tau and MAP1B is to “dynamically stabilize” microtubules and thus enable, among other things, mitochondrial transport in the cell. Therefore, we will only refer to the papers cited by the Reviewer.

The first study mentioned by the Reviewer was performed in cancer cells, and it demonstrates that overexpression of Tau was responsible for the interference with the attachment of cargoes to kinesin-based motors. Although the authors suggested that the Tau protein in the studied model was “in the range of endogenous MAP levels in the control cells and increases the amount of total MAP only about two-to threefold” it is hard to speculate whether this was the correct conclusion since several recent quantitative proteomic data show huge differences in the titer of Tau (and tubulin) in various cells. Tau overexpression is typical in neurodegenerative diseases. Numerous studies have shown that in neurons, Tau overexpression results in abnormally high levels of soluble Tau molecules what in turn, alters fast axonal transport, particularly in the anterograde direction. This alteration results in disruption of mitochondrial distribution, morphology and function, partially because microtubules lacking bound Tau are more prone to actions of MT depolymerizing agents; also the overexpression of Tau is responsible for the interference with trafficking of mitochondria in neurons (e.g., PMID: 18000878, 18798283, 21854751, 11901170).

In turn, the second paper cited by the Reviewer describes the effect of Tau and MAP1B knockout on mitochondria and tubulin in neurons, and it is very controversial since the authors achieved practically complete reduction of both studied proteins with any apparent effect on neuronal survival etc., which seems hardly possible taking into account their roles in growing axons. Nevertheless, event authors of the paper agreed that one of the roles of MAP1B and Tau is stabilization of MT. They suggested that MAP1B and Tau might regulate of direction of mitochondrial movement – depending on densities of the MAPs in spatial proximity to motor proteins. However, they did not mention how to reconcile the effect of the lack of the MAPs on microtubule structure with the acceleration of mitochondria trafficking.

  1. “The data with iFBP2 show an interphase microtubule network destabilisation effect and the presence of a subpopulation of shorter MT cytoskeleton similar to the low nocodazole resistant MTs as shown in Fig9C. The low nocodazole resistant MTs are known to be rich in post translationally modified alpha tubulin,  such as detyrosinated and acetylated alpha tubulin. The authors need to test if indeed is the case fo the iFBP2  treated cells. There is the possibility that the MT induced destabilization in the presence of iFBP2 may not be directly related to  inhibition of FBP2, especially since as depletion of FBP2 does not produce the same MT destabilization.”

Response: Reduction of FBP2 titer (both tetramers and dimers) does not recapitulate the effect of the chemically forced tetramerization of the enzyme in its inactive state. Because the information about the inhibitor-induced oligomerization of FBP2 in its inactive state appears to rise some difficulties in the interpretation of our data, we added some information to the Introduction chapter (page 1):

„In mammalian tissues, two different FBP isozymes are expressed: the liver FBP (FBP1) and muscle (FBP2) isozyme. They catalyze hydrolysis of fructose-1,6-phosphate to fructose-6-phosphate and inorganic phosphate. Both isozymes form homotetramers. In the presence of their allosteric inhibitors – AMP and NAD+, FBP1 and FBP2 tetramers adopt a similar, inactive T-state in which two upper subunits (the upper dimer) are slightly rotated in respect to the lower two subunits (the lower dimer). In the absence of the inhibitors, tetramers of both isozymes adopt the active R-state. But while the FBP1 tetramer is almost flat (for review see [7]), FBP2 adopts a unique cross-like structure in which the upper dimer is twisted about 90o in respect to the lower one [8]. FBP2 may also exist as the dimer which is fully active and is not inhibited by AMP and NAD+ since the mechanism of the allosteric inhibition requires the presence of the tetrameric conformation: binding of AMP to subunits within one dimer inhibits the catalytic sites in the second dimer. Importantly, both in the unique R-state of FBP2 and in its dimeric form, additional surfaces are exposed to the solution and hence, they may form new (as compared to FBP1 and also to the T-state of FBP2 tetramer) docking sites for binding partners (for review see [7]). “

  1. Deacetylation of α-tubulin reduces the efficiency with which motor proteins bind to microtubules and this could lead to mitochondria transport more or less similar to the defects observed by the authors (doi.org/10.1016/j.cub.2006.09.014). At the same time though, data strongly indicate that HDAC6 plays an important role in mitochondrial transport through its deacetylation effect on Miro1, a well-known partner in the Miro/Milton complex that links mitochondria to motor proteins. Deacetylation of Miro1 has been previously shown to decrease mitochondrial transport in axons (J Cell Biol 019 Jun 3;218(6):1871-1890 doi: 10.1083/jcb.201702187). Therefore, HDAC6 inhibition would increase the acetylated Miro1 and may restore the observed loss of motility of mitochondria following depletion or inhibition of FBP2. The authors may test this possibility.”

Response: The hypothesis of the Reviewer that FBP2 tetramerization in its inactive state may affect somehow HDAC is interesting, however, it is hard to see any direct connection between FBP2 inhibition and tubulin deacetylation except the apparent similarity of the effects.

  1. Since  iFBP2 induces tetramerization of FBP2 then where does the LPA tetramer-Phospo-tau complex colaclize? (Not on mitochondria not on MTs)”

Response: They probably colocalize in the cytosol. But the important thing is that they do colocalize and that it is correlated with MT destabilization.

  1. “Given the data presented in Fig 8 and the decrease in MT-mitochondria co-localization after FBP2 tetramerization, is in’t surprising that there is only 25% decrease in the velocity of Mitochondria displacement on Mts?”

Response: A decrease in MT-mitochondria colocalization is about 35% and the reduction in mitochondria velocity is about 25% thus, the values are very similar.

  1. “The presented data in figure 7 argue for enhanced interaction of tetrameric FBP2 with tau protein which also argues for the presence of tetrameric FBP2 in the cytoplasm. Why the iFBP2 induced FBP2 tetramers are not nuclear? Does FBP2 in iFBP2 treated cells all become nuclear since it become tetrameric?”

Response: Tetrameric FBP2 may be both cytosolic and nuclear. To be exported from nuclei it must dissociate into dimers (the putative nuclear export signal is located on the surface of interaction between the upper and the lower dimers within the tetramer; PMID: 29383170). Although one cannot exclude that the tetramer can be exported from nucleus using e.g., the piggy-back mechanism.

  1. Where are the data supporting the statement in lines 298-301. The dimer (cytoplasmic) to tetrameric (nuclear) in untreated cells can be determined but the dimer to tetramer ratio  in cytoplasm it is impossible to determine by the methods presented here.”

Response: We stated: “Apparently, while silencing of FBP2 expression reduced the overall amount of FBP2 protein, it did not sufficiently decrease dimer-tetramer ratio in the cytoplasm to alter the MT cytoskeleton architecture.” and now we understand that it was unclear. In the corrected version of the manuscript, the sentence reads: “Most likely, in contrast to forced tetramerization of FBP2 molecules, partial silencing of FBP2 expression reduced the overall amount of FBP2 protein, but it did not decrease the dimer-tetramer ratio in the cytoplasm and hence, no alterations in the MT cytoskeleton architecture were observed”.

 Minor points

The manuscript needs to be edited by a native English speaking editor since there is a lot of English language editing is required. Below I pinpoint some of the necessary editions.

Response: We agree that we should have re-read the manuscript more carefully, and noticed the errors, and we are grateful to the Reviewer for pointing them out, especially the incorrect citation of a figure and an axis caption. We hope that the current version of the manuscript is free from errors. Since in several cases, the two Reviewers had different opinions about necessary corrections (whether to change "what" to "and" or to "which"), we consulted our English colleague and corrected the manuscript according to his suggestions. At the same time, we would like to explain why we used some phrases:

Line 36 replace ‘such’ to ‘certain’

Response: We used the phrase "such conditions" because it refers to the conditions mentioned in the previous sentence.

Line 273 edit ‘Quite a some time ago, …’ to ‘We have previously reported…’

Response: By using the phrase we meant to emphasize that our search for the proteins took place many years ago. However, we agree that it may be too informal and thus, we changed it to “Previously”.

From the material and Methods section as well from the result section and figure legend it is not stated after how many hours or days following RNAi depletion the data are presented.

Response: We are very sorry for this oversight. We clarified this in the subchapter 2.4. Silencing of FBP2 expression: “A stable line of shRNA expressing HL-1 cells was established using puromycin dihydrochloride (15 µg/ml).”

We hope that our amendments of the manuscript will meet with the Reviewer’s approval.

Round 2

Reviewer 2 Report

In the manuscript, the phrase ‘cluster of mitochondria’ is mentioned first in the context of the VAL115Met mutant in lines 67-71.  In the next paragraph the authors state ‘Therefore, we sought to clarify the contribution of FBP2 to this clustering of mitochondria in HL-1 cardiomyocytes…’ As the authors state in their response to the reviewer letter and if I am not mistaken the HL-1 cardiomyocytes express the wild type and not the mutant FBP2 protein, therefore clustering of mitochondria is not normally observed in cells. Then, if it is not the VAL115Met,  which of their earlier observations prompted them to investigate the involvement of FBP2 in mitochondrial motility in the cell? The authors need to clarify to the readers what is their definition of ‘mitochondria clusters’ and wether what they describe in lines 75-79 is the same as ‘mitochondria clusters’. The authors should make an effort to understand that their paper is addressed to readers that they may not be experts on mitochondria physiology (loss of mitochondrial motility = mitochondrial clustering?).

It is still puzzling how a a 35% reduction of protein levels leads close to 62% loss of enzymatic activity (Fig1). As if  there is a critical concentration for the enzyme that below this concentration the enzyme is less active.  Is there an explanation for this huge loss of activity given the low protein depletion? Is there monomer dimer equilibria in cells, and does the activity of the monomer differ from the dimer? Because if there is, then the 35% reduction of protein levels may shift the equilibrium to monomeric FBP2 and therefore loss of activity (?).

The authors in the response letter and in the manuscript state ‘that the silencing - that reduced the overall amount of FBP2 protein but apparently, did not change the dimer-tetramer ratio in the cytoplasm – did not alter significantly mitochondrial length, mitophagy and the MT cytoskeleton architecture.’ Where are the data showing that silencing, did not change the dimer-tetramer ratio in the cytoplasm and how did they measure the ratio in the cytoplasm?

I would agree with the authors that the iFBP2 is a commercially available inhibitor and its proprieties and effects on FBP2 nucleo-cytoplasmic localization has been demonstrated in several papers, also published previously by their group.  Frankly, I do not know anything about iFBP2 and in the current context of the paper, as a reviewer and as a naive reader, it is not clear whether the drug induced tetramer is cytoplasmic or nuclear. The authors state that FBP2 needs to be in a dimeric form to exit the nucleus. According to the information presented in the manuscript, the tetramer is nuclear and therefore I assume the drug induced tetramer is nuclear and therefore the drug may deplete the available FBP2 dimer from the cytoplasm. The observed  reduced interaction of FBP2 with mitochondria in the presence of iFBP2 may not be because it cannot bind to mitochondria but because all protein is in tetrameric form in the nucleus. The LPA complex detected in the cytoplasm after treatment with iFBP2 may not be necessarily the tetrameric form of FBP2. If the drug induced tetramer is not nuclear then  why not ? Is the nuclear localisation signal, if it exists, masked in the tetramer?   Where in the manuscript all these possibilities discussed and/or eliminated? All the questions raised here are due to the fact there is background information and data which are missing, leading the reader to draw conclusions, like I state them here in the review, that may not be valid but with the presented data and background information are certainly justified.

The authors have previously characterised a FBP2 mutant  (L190G; new ref:9)  that retains its activity and fails to form tetramers. Therefore they are in the best position to show that cells expressing the mutant are iFBP2 resistant and expression of this mutant rescues the mitochondrial motility defects induced by iFBP2 and FBP2 tetramerization. 

The argument that the structure of the iFBP2 inhibitor differs from nocodazole  and therefore and cannot act as nocodazole is not valid since a lot of different structurally distinct tubulin inhibitors bind to the same nocodazole binding site  and other tubulin sites and weak microtubule  inhibitors give similar phenotypes as the low nocodazole concentrations. 

There is no doubt of the MT stabilisation roles of Tau and MAP1B and there is plenty of evidence that tau over expression (possibly at no physiological levels) interferes with mitochondrial transport.None the less,   loss of MT bound tau should facilitate mitochondrial transport and not inhibit it, as it is shown here by the tetramerization of FBP2 protein. The authors thesis that loss of FBP2 activity and/or tetramerization of FBP2 leads to MT dynamics changes and affect mitochondrial-MT interaction,  as an observation is important and that is why it is critical to eliminate other possibilities such as the nonspecific effects of iFBP2 inhibitor on MT dynamics. The authors choose not to address this possibility arguing that the inhibitor has been studied previously and is commercially  available and does not look like nocodazole.

Line 41: edit to exists as a dimer

Author Response

“In the manuscript, the phrase ‘cluster of mitochondria’ is mentioned first in the context of the VAL115Met mutant in lines 67-71.  In the next paragraph the authors state ‘Therefore, we sought to clarify the contribution of FBP2 to this clustering of mitochondria in HL-1 cardiomyocytes…’ As the authors state in their response to the reviewer letter and if I am not mistaken the HL-1 cardiomyocytes express the wild type and not the mutant FBP2 protein, therefore clustering of mitochondria is not normally observed in cells. Then, if it is not the VAL115Met,  which of their earlier observations prompted them to investigate the involvement of FBP2 in mitochondrial motility in the cell? The authors need to clarify to the readers what is their definition of ‘mitochondria clusters’ and wether what they describe in lines 75-79 is the same as ‘mitochondria clusters’. The authors should make an effort to understand that their paper is addressed to readers that they may not be experts on mitochondria physiology (loss of mitochondrial motility = mitochondrial clustering?).”

Response: In the cell, mitochondria exist in two forms: as isolated particles and as filamentous structures which often interconnect to form extended networks. They dynamically change their shapes and distribution in the cell, through fission, fusion and motility along microtubules. In cells we are studying – e.g., fibroblasts and HL-1 cells, mitochondria (regardless of their shape) are more or less evenly distributed throughout the cell. However, under some circumstances – e.g., in fibroblasts carrying the Val115Met mutation of FBP2 – this distribution changes and the organelles aggregate (cluster) in  the central regions of cells, rather than form extended networks. This aggregation might be a result of diminished motility of the organelles. The Val115Met FBP2 mutation influences subcellular localization of the mutant FBP2. Notably, the mutant does not co-localize with mitochondria. Therefore, the disturbance in FBP2-mitochondria interaction correlates with changes in normal distribution of the organelles. This might suggest that perturbation of the FBP2-mitochondria interaction leads to changes in their motility. In a healthy cell, such an interaction might be disturbed by tetramerization of FBP2 (by physiological effectors – AMP or NAD+, or by a synthetic compound mimicking their action on FBP2). Thus, we decided to test if forced tetramerization of FBP2 may influence mitochondrial motility in the cell.

We agree with the Reviewer's opinion that our description in the original text was indeed insufficient, so we extended it to include the above-mentioned information (Introduction, page 2; the new text italicized):

“Recently, we have found that a novel remitting leukodystrophy was associated with a Val115Met variant of FBP2 [11]. In fibroblasts carrying the variant, FBP2 was unable to co-localize with mitochondria which correlated with a disturbance of mitochondrial network and increase in ROS production. In particular, in the majority of the cells, mitochondria appeared to aggregate in  the central region of cells, rather than form extended networks as in healthy cells. This aggregation could be a result of diminished motility of mitochondria along microtubules. This in turn, might suggest that the perturbation of the Val115Met FBP2-mitochondria interaction contributed to the changes in motility of the organelles. In healthy cells, such an interaction might be disturbed by induction of tetramerization of FBP2 (e.g., by its physiological effectors – AMP or NAD+) [9].

Therefore, we sought to clarify the contribution of FBP2 to the aggregation of mitochondria in HL-1 cardiomyocytes – the cells that express only the FBP2 isozyme, and in which the pathways regulating intracellular behavior of the protein was best studied.”

“It is still puzzling how a a 35% reduction of protein levels leads close to 62% loss of enzymatic activity (Fig1). As if  there is a critical concentration for the enzyme that below this concentration the enzyme is less active.  Is there an explanation for this huge loss of activity given the low protein depletion? Is there monomer dimer equilibria in cells, and does the activity of the monomer differ from the dimer? Because if there is, then the 35% reduction of protein levels may shift the equilibrium to monomeric FBP2 and therefore loss of activity (?).

The authors in the response letter and in the manuscript state ‘that the silencing - that reduced the overall amount of FBP2 protein but apparently, did not change the dimer-tetramer ratio in the cytoplasm – did not alter significantly mitochondrial length, mitophagy and the MT cytoskeleton architecture.’ Where are the data showing that silencing, did not change the dimer-tetramer ratio in the cytoplasm and how did they measure the ratio in the cytoplasm?”

Response: As the Reviewer suggested, lower activity of FBP2 may result from its dissociation into monomers which are inactive. However, monomers do not appear to exist in the cell. This is because the substrate/product of FBP2 reaction stabilizes dimers (e.g., PMID: 11257504) and the cellular titer of fructose-1,6-bisphosphate and fructose-6-phosphate is many times higher than their Kd for FBP2. The titer of the substrate is about 30 µM, sometimes (e.g., in cancer cells) significantly higher than Kd (or Km, that corresponds to Kd in Michealis-Menten kinetics) which is between 1 µM and 5 µM (both for FBP2 and FBP1). The monomers also cannot affect the measurements in vitro since the substrate concentration in a assay is 200 µM.

For the consistency of the text, we would prefer not to discuss kinetic properties of FBP2, in the present manuscript.

We did not measure the cytoplasmic ratio of dimers to tetramers after the silencing and induced tetramerization because, technically, it is not possible. Obviously, a decrease in a protein titer may reduce the formation of higher oligomeric complexes but when the reduction is relatively small (e.g., about two-fold as in our case) the changes in the ratio of the complex to interactants are negligible. We demonstrated that using analytical ultracentrifugation just for FBP2 (there were no significant differences in ratio of various FBP2 oligomeric states when we used different FBP2 concentrations: 0.1, 0.5 and 1 mg/ml; Wiśniewski et al. PMID: 29383170).

All techniques based on immunoassays, such as IF or WB, are semiquantitative methods only; and results our measurements are just good examples of that: standard deviations are relatively high due to both biological variability of the cells and uncertainty of the measurement. Enzyme activity measurement is obviously a more analytical method but it also contains some uncertainty e.g., resulting from the presence of various titers of known physiological effectors and/or the presence of yet unknown ligands. Thus, one should not compare experimental values obtained using two different techniques, although both of them provide information about changes.

“I would agree with the authors that the iFBP2 is a commercially available inhibitor and its proprieties and effects on FBP2 nucleo-cytoplasmic localization has been demonstrated in several papers, also published previously by their group.  Frankly, I do not know anything about iFBP2 and in the current context of the paper, as a reviewer and as a naive reader, it is not clear whether the drug induced tetramer is cytoplasmic or nuclear. The authors state that FBP2 needs to be in a dimeric form to exit the nucleus. According to the information presented in the manuscript, the tetramer is nuclear and therefore I assume the drug induced tetramer is nuclear and therefore the drug may deplete the available FBP2 dimer from the cytoplasm. The observed  reduced interaction of FBP2 with mitochondria in the presence of iFBP2 may not be because it cannot bind to mitochondria but because all protein is in tetrameric form in the nucleus. The LPA complex detected in the cytoplasm after treatment with iFBP2 may not be necessarily the tetrameric form of FBP2. If the drug induced tetramer is not nuclear then  why not ? Is the nuclear localisation signal, if it exists, masked in the tetramer?   Where in the manuscript all these possibilities discussed and/or eliminated? All the questions raised here are due to the fact there is background information and data which are missing, leading the reader to draw conclusions, like I state them here in the review, that may not be valid but with the presented data and background information are certainly justified.”

Response: It was not our intention to suggest that all tetrameric FBP2 molecules are located in the nuclei. It would be possible only if the tetramerization was the signal for nuclear transport of FBP2 (e.g., if FBP2 had the spatial epitope Nuclear Localization Signal and all four subunit contributed to its formation) and all the tetrameric FBP2 molecules were transported to nuclei. We do not have any evidence supporting such a possibility and thus, we do not discuss it. NLS of FBP2 is exposed to solution on the surface of each monomer (PMID: 19626708), independently to the oligomeric state, and because the drug is a low molecular compound it probably may stimulate the tetramerization in all cellular compartments.

“The authors have previously characterised a FBP2 mutant  (L190G; new ref:9)  that retains its activity and fails to form tetramers. Therefore they are in the best position to show that cells expressing the mutant are iFBP2 resistant and expression of this mutant rescues the mitochondrial motility defects induced by iFBP2 and FBP2 tetramerization.”

Response: Seemingly, such a procedure could demonstrate the rescue. The problems with a compensation of the function by this mutant are, however, that it is constantly active and it may interact with several other proteins (not only mitochondrial) than WT FBP2. In HL-1 cardiomyocytes, WT FBP2 is expressed at a relatively low level (as compared to e.g., glycolytic or ribosomal proteins). Thus, it is hardly possible to keep the level of the overexpressed mutant FBP2 on the level similar to the native protein. Therefore, the overexpression of the mutant FBP2 would not rescue the lack of dimeric FBP2 but in fact, create a new type of cell in which FBP2 level would be strongly elevated (mainly due to the expression of the dimeric form) and of different metabolism (e.g. accelerated gluco-/glyconeogensis because dimeric FBP2 is fully active and insensitive to allosteric inhibition). The higher titer of FBP2 molecules (both dimers and tetramers) is also assumed to affect the rate of glycolysis, mitochondrial oxidation and mitochondrial biogenesis, as it has been shown that FBP2 (and FBP1) affects the levels of e.g., Hif1α and NF-κB (we described it in the Introduction: “This is because it has been shown that FBP may regulate not only glucose/glycogen synthesis from precursors of carbohydrates but it may also influence – via interactions with a set of proteins (e.g., mitochondrial VDAC, ANT, and ATP synthase, CAMK2, and transcription factors HIF1α and NF-κB) – cell cycle-dependent events, mitochondria biogenesis and polarization of their membranes, expression of glycolytic enzymes, induction of synaptic plasticity and even cancer progresssion [1-6]”

As a matter of fact, it is a common problem of numerous studies in which researches try to “rescue” the lack an enzyme – one does not know what sort of cells they have after the overexpression.

“The argument that the structure of the iFBP2 inhibitor differs from nocodazole  and therefore and cannot act as nocodazole is not valid since a lot of different structurally distinct tubulin inhibitors bind to the same nocodazole binding site  and other tubulin sites and weak microtubule  inhibitors give similar phenotypes as the low nocodazole concentrations. 

There is no doubt of the MT stabilisation roles of Tau and MAP1B and there is plenty of evidence that tau over expression (possibly at no physiological levels) interferes with mitochondrial transport.None the less,   loss of MT bound tau should facilitate mitochondrial transport and not inhibit it, as it is shown here by the tetramerization of FBP2 protein. The authors thesis that loss of FBP2 activity and/or tetramerization of FBP2 leads to MT dynamics changes and affect mitochondrial-MT interaction,  as an observation is important and that is why it is critical to eliminate other possibilities such as the nonspecific effects of iFBP2 inhibitor on MT dynamics. The authors choose not to address this possibility arguing that the inhibitor has been studied previously and is commercially  available and does not look like nocodazole.”

Response: We are not sure how the loss of  MT-bound Tau could facilitate mitochondrial transport when at the same time, the disturbance of MT structure and the reduction of MT-mitochondria colocalization are observed.

We would also prefer not to repeat our comments on the publications that the Reviewer provided previously in support of his statement.

We agree that there is a variety of tubulin polymerization inhibitors other than nocodazole, and it cannot be excluded that any compound may affect tubulin structure even though iFBP2 and nocodazole binding to the same site appears unlikely.

The inhibitor has been tested in hepatoma cells, which do not express FBP2 but solely FBP1, and the researchers did not observe any negative effects of the inhibitor, except the gluconeogenesis inhibition (von Goldern et al. PMID: 16442285).

Establishing the titer of the inhibitor we tested the effect of its 2, 5 and 10 µM concentration on mitochondria polarization and structure of their network, and we found that the effects of 2 µM iFBP2 were weak and inconclusive, while the effects of 5 and 10 µM iFBP2 were indistinguishable, and thus, we used 5 µM iFBP2. This concentration is about 5 to 3 times higher than IC50 determined for FBP1 and FBP2 (the data for FBP2 have not been published yet).

It would be quite surprising if IC50 for FBP2 was similar to the titer of the inhibitor necessary to influence microtubule structure but, obviously, we cannot exclude such a coincidence.

To the newest version of the manuscript, we added information about the possibility that the inhibitor can act as a tubulin-interacting molecule (Subchapter 3.2, page 8):

“We cannot completely exclude the possibility that iFBP2 might directly affect the structure of MT, however, such effect was not observed in a study using hepatoma cells [18].”

On the other hand we did not suggest that the loss of FBP2 activity was related to changes in MT/mitochondria-MT interaction. We suggested that the oligomeric state of FBP2 was responsible for the observed changes.

“Line 41: edit to exists as a dimer”

Response: We changed this accordingly.